# MIC26 and MIC27 are bona fide subunits of the MICOS complex in mitochondria and do not exist as glycosylated apolipoproteins

Melissa Lubeck[1], Nick H. Derkum[1], Ritam Naha[1], Rebecca Strohm[1], Marc D. Driessen[2], Bengt-Frederik Belgardt[3,4], Michael Roden[4,5,6], Kai Stühler[2,7], Ruchika Anand[1], Andreas S. Reichert[1], Arun Kumar Kondadi[1] *

1 Medical Faculty and University Hospital Düsseldorf, Institute of Biochemistry and Molecular Biology I, Heinrich Heine University Düsseldorf, Düsseldorf, Germany, 2 Medical Faculty and University Hospital, Institute of Molecular Medicine, Protein Research, Heinrich Heine University Düsseldorf, Düsseldorf, Germany, 3 Institute for Vascular and Islet Cell Biology, German Diabetes Center (DDZ), Leibniz Center for Diabetes Research at Heinrich Heine University, Düsseldorf, Germany, 4 German Center for Diabetes Research (DZD e.V.), Partner Düsseldorf, Neuherberg, Germany, 5 Medical Faculty and University Hospital Düsseldorf, Department of Endocrinology and Diabetology, Heinrich Heine University, Düsseldorf, Germany, 6 Institute for Clinical Diabetology, German Diabetes Center, Leibniz Center for Diabetes, Heinrich Heine University, Düsseldorf, Germany, 7 Molecular Proteomics Laboratory, BMFZ, Heinrich Heine University Düsseldorf, Düsseldorf, Germany

* kondadi@hhu.de

**Data Availability Statement:** All relevant data are within the manuscript and its Supporting information files.

## Abstract

Impairments of mitochondrial functions are linked to human ageing and pathologies such as cancer, cardiomyopathy, neurodegeneration and diabetes. Specifically, aberrations in ultra-structure of mitochondrial inner membrane (IM) and factors regulating them are linked to diabetes. The development of diabetes is connected to the 'Mitochondrial Contact Site and Cristae Organising System' (MICOS) complex which is a large membrane protein complex defining the IM architecture. MIC26 and MIC27 are homologous apolipoproteins of the MICOS complex. MIC26 has been reported as a 22 kDa mitochondrial and a 55 kDa glyco-sylated and secreted protein. The molecular and functional relationship between these MIC26 isoforms has not been investigated. In order to understand their molecular roles, we depleted *MIC26* using siRNA and further generated *MIC26* and *MIC27* knockouts (KOs) in four different human cell lines. In these KOs, we used four anti-MIC26 antibodies and con-sistently detected the loss of mitochondrial MIC26 (22 kDa) and MIC27 (30 kDa) but not the loss of intracellular or secreted 55 kDa protein. Thus, the protein assigned earlier as 55 kDa MIC26 is nonspecific. We further excluded the presence of a glycosylated, high-molecular weight MIC27 protein. Next, we probed GFP- and myc-tagged variants of MIC26 with anti-bodies against GFP and myc respectively. Again, only the mitochondrial versions of these tagged proteins were detected but not the corresponding high-molecular weight MIC26, suggesting that MIC26 is indeed not post-translationally modified. Mutagenesis of predicted glycosylation sites in MIC26 also did not affect the detection of the 55 kDa protein band. Mass spectrometry of a band excised from an SDS gel around 55 kDa could not confirm the presence of any peptides derived from MIC26. Taken together, we conclude that both

**Funding:** 1) Funded by the Deutsche Forschungsgemeinschaft (DFG) Graduiertenkolleg VIVID RTG 2576 to ASR (www.dfg.de) 2) DFG Grant KO 6519/1-1 to AKK 3) DFG Grant AN 1440/3-1 to RA The funders had no role in study design, data collection and analysis, decision to publish, or preparation of the manuscript.

**Competing interests:** The authors have declared that no competing interests exist.

MIC26 and MIC27 are exclusively localized in mitochondria and that the observed phenotypes reported previously are exclusively due to their mitochondrial function.

## Introduction

Mitochondria are essential double-membrane-enclosed organelles required for a myriad key cellular functions including ATP conversion, iron-sulfur cluster biogenesis, programmed cell death, calcium buffering, vital metabolic processes and inflammatory responses. The mitochondrial inner membrane (IM) is spatially heterogenous and composed of the inner boundary membrane (IBM), parallel to the outer membrane (OM), and the cristae membrane (CM) extending as invaginations towards the mitochondrial matrix. CMs house the electron transport chain (ETC) complexes [1,2] and are responsible for ATP generation. Cristae exist in a variety of shapes and sizes depending on the cell, tissue type and bioenergetic status of the cell [3,4]. Aberrant cristae are found in a wide variety of pathologies like diabetic cardiomyopathy, neurodegeneration and mitochondrial myopathy [5–7]. The IBM and CM are separated by distinct pore-like structures which are around 25 nm in diameter and termed crista junctions (CJs) [8,9]. CJs were proposed as a diffusion barrier for various metabolites, membrane and soluble proteins [10–13]. In congruence, it was shown that there was differential localisation of various proteins in the IM: The IBM predominantly contains proteins involved in mitochondrial fusion and protein import while the CMs are enriched with proteins belonging to mitochondrial protein synthesis, iron-sulfur cluster biogenesis and ETC complexes [1,2]. In fact, the CJs give rise to heterogenous membrane potential of the individual cristae [14]. Therefore, CJs are an important structural and functional constituent of the IM.

The MICOS (Mitochondrial contact site and cristae organising system) complex is a multi-protein complex comprising of seven *bona fide* proteins which resides at the CJs. The MICOS complex was discovered in *Saccharomyces cerevisiae* by independent groups [15–17]. The proteins of the MICOS complex include two subcomplexes: MIC60/MIC25/MIC19 and MIC10/MIC26/MIC27 which are bridged by MIC13 [18,19] using its conserved GxxxG and WN motifs [20]. The MICOS complex present in the IM is in turn connected via MIC60 to SAM50 present in the OM through MIC19 contributing to the formation of a 2 mega Dalton MIB (Mitochondrial intermembrane space bridging) complex [21,22]. Thus, the MIB complex mediates the IM and OM contacts. The MICOS complex correlates with the presence of cristae. In fact, a very recent study showed that Mic60 is involved in the development of intracytoplasmic membranes in alphaproteobacteria [23]. Mic60 is conserved from alphaproteobacteria to virtually all eukaryotes, whereas Mic10 is widely distributed in the later domain of life [24,25]. However, there is a difference w.r.t Mic10 and Mic60 in organisms which are opisthokonts and not, like *Trypansoma brucei*. TbMic60 lacks a conserved mitofilin domain while there are two TbMic10 proteins inconsistent with opisthokonts [26]. Other MICOS proteins display inconsistent phylogenetic distribution. For instance, MIC26 and MIC27 are paralogs which are only present in opisthokonts [24,25]. The MICOS complex proteins play a wide range of roles in bending the IM, formation of contact sites, protein import, mtDNA organisation, lipid metabolism, cristae dynamics, mitochondrial motility, integrity of the respiratory chain super complexes and the $F_1F_O$-ATP synthase [3,17,27–34]. Consequently, it is well established that proteins of the MICOS complex are associated with various pathologies [35,36]. Mutations in *MIC60* were shown to be present in Parkinson's disease patients [37]. A recent report showed that homozygous mutations in *MIC60* led to developmental encephalopathy, optic atrophy, dystonia and nystagmus [38]. *MIC13* mutations cause mitochondrial

encephalopathy and hepatic disorders [39]. It is also known that mutations in *MIC26* cause mitochondrial myopathy, cognitive defects and lactic acidosis [40]. Further, there is an intricate connection between MICOS proteins and diabetes. Interfibrillary cardiac mitochondria showed a decreased amount of MIC60 in type I diabetic hearts indicating mitochondrial dysfunction contributes to diabetic cardiomyopathy [41]. When control and transgenic mice, overexpressing *Mic60*, were treated with streptozotocin to induce diabetes, the transgenic mice displayed improved mitochondrial ultrastructure and cardiac contractile functions compared to control mice [42]. Hence, increased amount of MIC60 is beneficial to overcome at least some cellular abnormalities in diabetes. Regarding other MICOS proteins, hearts of diabetic patients showed an upregulation of *MIC26* mRNA pointing to a close link between diabetes and the mitochondrial apolipoprotein MIC26 [43]. Overexpression of *Mic26* in murine liver resulted in steatosis where the effects were more pronounced in mice fed with high-fat diet compared to normal diet [44]. Further, triglyceride content was increased in liver upon *Mic26* overexpression when mice were fed with normal as well as high-fat diet while the effects were pronounced with high-fat diet. In addition, the mRNA expression of genes involved in fatty acid metabolism was altered.

Complexome profiling of purified bovine mitochondria resulted in the identification of two homologous apolipoproteins, MIC26/APOO and MIC27/APOOL, as a part of mammalian MICOS complex [45]. Accordingly, an altered mitochondrial ultrastructure and function was observed in cells depleted for MIC26 and MIC27 [31,46] as well as in cells including cardiac myoblasts overexpressing *MIC26* and *MIC27* [45,47]. Other studies have demonstrated the structural, functional and pathological importance of MIC26 localized to the mitochondria [31,40,46,48,49] which we designate as $MIC26_{22kDa}$ throughout this manuscript. Before MIC26 was shown to be present in mitochondria as a 22 kDa protein [46–48], it was observed that the MIC26 apolipoprotein was O-glycosylated and present as a 55 kDa protein, termed $MIC26_{55kDa}$ from now on [43]. In fact, MIC26 was initially found as a protein of unknown function while studying cardiac transcriptome in dogs fed with hypercaloric high-fat diet in an obesity-related hypertension model [50]. This protein of unknown function which contained a couple of amphipathic α-helices, a common feature of apolipoprotein family, was enriched in high-density lipoproteins and therefore identified as an apolipoprotein and given the name ApoO in chronological succession [43]. A protein band at 55 kDa was observed in human, dog and murine serum as well as in human auricle and HepG2 cells. However, the recombinant protein resulted in a size of 22 kDa. In order to explain the discrepancy between the observed 55 kDa and the 22 kDa recombinant protein, the authors predicted MIC26 is glycosylated and found that $MIC26_{55kDa}$ was sensitive to chondroitinase ABC (cABC) deglycosylation releasing a maximum of 50% of the protein at 22 kDa [43]. This was validated by preventing the MIC26 glycosylation by using *p*-nitrophenyl-*β*-D-xyloside which increased the 22 kDa protein in modest amounts. Following this characterisation, it was deemed to be an original glycoprotein. $MIC26_{55kDa}$ was increased around two-fold in plasma of patients suffering from acute coronary syndrome [51]. When HepG2 cells were depleted for MIC26, microarray analysis revealed genes involved in fatty acid metabolism and inflammatory responses were differentially expressed [52]. Transgenic overexpression of *Mic26* mice fed with a high-fat diet resulted in alterations of mitochondrial ultrastructure, cardiac and metabolic functions in that there was an enhanced accumulation of diglycerides in *Mic26* overexpressing mice [47]. Other reports immediately after this showed that $MIC26_{22kDa}$ was conclusively localized to the mitochondria [46,48]. Hence, a dual role of MIC26 as a mitochondrial as well as glycosylated protein emerged into prominence. However, studies in chicken after the discovery of the mitochondrial $MIC26_{22kDa}$ still focussed exclusively on the so-called $MIC26_{55kDa}$ secreted version [53,54]. Therefore, it is important to study both the $MIC26_{22kDa}$ and $MIC26_{55kDa}$

together. Although it is clear that increased amounts of MIC26 in mice lead to defects at the cellular and organismal level [47], the significance and relevance of the $MIC26_{55kDa}$ is not clearly understood. In fact, to the best of our knowledge, the 55 kDa protein of MIC26 was not studied in *MIC26* knockouts (KOs). In addition, MIC26 was not found to be a part of high-density lipoprotein (HDL)-associated proteins in a proteomic study where other major apolipoproteins could be detected [55]. Therefore, there are considerable inconsistencies regarding $MIC26_{55kDa}$. In this manuscript, we focussed to solve this conundrum. In fact, in light of multiple experiments, we consider that the recognised 55 kDa protein band is nonspecific and must be treated with caution in future research endeavours.

## Materials and methods

### Cell culture and downregulation using siRNA

HEK293 and HeLa cells were maintained in 1 g/L glucose DMEM (PAN-Biotech) supplemented with 10% fetal bovine serum (FBS, Capricorn Scientific), 2 mM GlutaMAX (Gibco), 1 mM sodium pyruvate (Gibco) and penstrep (PAN-Biotech, penicillin 100 U/mL and 100 µg/mL streptomycin). HepG2 cells were cultured in 1 g/L glucose DMEM (PAN-Biotech) supplemented with 10% FBS (Capricorn Scientific), 2 mM GlutaMAX (Gibco) and penstrep (PAN-Biotech). C2C12 cells were cultured in 4.5 g/L DMEM (Pan-Biotech) supplemented with 10% FBS (Capricorn Scientific) and penstrep (PAN-Biotech), whereas HAP1 cells were maintained in Iscove's modified DMEM medium (IMDM, Sigma-Aldrich) supplemented with 20% FBS, 2 mM GlutaMAX (Gibco) and penstrep (PAN-Biotech). All the above-mentioned cells were grown at 37˚C supplied with 5% $CO_2$. For downregulation of *MIC26*, the cells were transfected with Lipofectamine™ RNAiMAX reagent (Invitrogen) according to the manufacturer's protocol. Downregulation was done for 48 h with 10 nM of siRNA. Negative control medium GC duplex (Invitrogen, cat no: 462001) was used as control along with following siRNA sequences (Thermo Fischer Scientific):

1. *MIC26* # 1 (ID30917 targeting exon 6 and 7): 5'-GGAUAUAUAGUCAUAGAAGTT-3'; 5'-CUUCUAUGACUAUAUAUCCTC-3'

2. *MIC26* # 2 (ID127368 targeting exon 3 and 4): 5'-GGAAACGUACUCCCAAACUTT-3'; 5'-AGUUUGGGAGUACGUUUCCTG-3'

3. *MIC26* # 3 (ID127366 targeting exon 2): 5'-CCUCCCAAAAAUUCCGUGATT-3'; 5'-UCACGGAAUUUUUGGGAGGTG-3'

### Generation of *MIC26* and *MIC27* KOs using the CRISPR/Cas method

*MIC26* and *MIC27* KOs in HepG2, HeLa and HEK293 cells were generated using the double nickase method. This CRISPR/Cas KO system consists of two plasmids where each plasmid contained a sgRNA sequence and a reporter (either GFP or puromycin resistance) for positive selection of cells. The plasmid mix was commercially available (Santa Cruz Biotechnology): MIC26: double nickase plasmid sc-413137-NIC and MIC27 double nickase plasmid sc-414464-NIC. Cells were seeded on 35 mm dishes and transfected one day later with 1 µg of the respective plasmid mix using GeneJuice (Novagen) reagent according to the manufacturers protocol. Two days after transfection, single cells were sorted by FACS (Beckman Coulter) using GFP fluorescence into a 96-well plate (Greiner) containing respective conditioned media to obtain homogeneous KO cell populations from single cells. HAP1 *MIC26* and *MIC27* KO cells were custom-made by Horizon (UK) as described before [31].

## Molecular cloning/Plasmids & Constructs

MIC26-GFP was generated by cloning human MIC26 in to the pEGFP-N1 vector, using the Kpn1 and Age1 restriction enzymes which was followed by ligation. MIC26[S34A]-GFP, MIC26[S41A]-GFP and MIC26[S50A]-GFP mutant variants of MIC26-GFP were generated using the Q5 site-directed mutagenesis kit (New England Biolabs). pMSCV-MIC26 plasmid, generated in a previous study [31], was used as a template for making pMSCV-MIC26-myc plasmid using the Q5 site-directed mutagenesis kit (New England Biolabs) according to manufacturer's protocol with the following primers:

1. myc-tag forward:
   5'-GAACAAAAACTCATCTCAGAAGAGGATCTCTAGCGAATTCTACCGGGTAG-3'

2. myc-tag reverse:
   5'-CCCAGATCCCTTAGTTCCAGGTGAATTCTTCACA-3'

For generating triple mutant MIC26[S34A/S41A/S50A]-myc plasmid, insertion of myc-tag and triple amino acid mutations from serine to alanine (S34, S41 and S50) were generated using the following primers:

3. MIC26-S34/41/50A forward:
   5'-CTACTCAG−TTCCTGAGGGTCAAGCGAAGTATGTGGAGGAGGCA-3'

4. MIC26-S34/41/50A reverse:
   5'-AGTGC−AAGCTCATCAACCTTCACGGCATTTTTGGGAGGTGAGTC-3'

In order to express MIC26-GFP and MIC26-myc variants, HEK293 cells were seeded onto 35 mm plates and transfected with 1 μg of corresponding plasmid using GeneJuice (Novagen) reagent according to the manufacturer's protocol and grown for 48 h until harvest.

## SDS gel electrophoresis and Western Blotting

Cellular proteins were harvested from a corresponding number of cells after washing them three times with 2 mL DPBS (PAN-Biotech) followed by scraping and resuspending the cells in an appropriate volume of lysis buffer (210 mM mannitol, 70 mM sucrose, 1 mM EDTA, 20 mM HEPES, 1 x protease inhibitor (Sigma-Aldrich)) or RIPA buffer (150 mM NaCl. 0.1% SDS, 0.05% DOC. 1% Triton-X-100. 1 mM EDTA, 1mM Tris, pH 7.4, 1 x protease inhibitor (Sigma-Aldrich)). Following the incubation of cells for 10 min on ice, they were mechanically disrupted using a 20G canula by repetitive strokes. In order to harvest secreted proteins, cells were seeded into 35 mm plates containing 2 mL standard growth medium. After 24 h incubation, cells were carefully washed three times with 2 mL DPBS (PAN-Biotech) and grown for another 24 h in the corresponding medium lacking FBS. The culture medium was collected and separated from detached cells by centrifugation for 5 min at 1000g and 4˚C. Next, proteins in cell culture media were precipitated by the addition of 10% trichloroacetic acid (TCA) for 40 min on ice. The precipitated proteins were pelleted by centrifugation for 10 min at 16,000 g, 4˚C and washed twice with 2 mL ice-cold acetone. The pellet was dried at room temperature, dissolved in 4 M Urea, 1 mM DTT, 2% SDS and heated for 5 min at 95˚C. For subcellular fractionation, nuclear fraction was removed from cell lysates, obtained by mechanical disruption, upon centrifugation for 5 min at 500g and 4˚C. An aliquot of the supernatant was taken as total fraction. Next, the mitochondrial fraction was pelleted by centrifugation at 7000g and 4˚C for 10 min. Mitochondrial fraction was washed three times and sequentially resuspended in lysis buffer. The supernatant contained the ER/Golgi and cytosolic fraction. For murine tissue lysates, 20 mg of tissue was homogenized, in Precellys Soft Tissue homogenizing CK14

tubes, with 300 μL RIPA buffer in a Precellys 24 homogenizer at 5000 rpm for 10 sec. Further, solubilized proteins were separated from the tissue debris by centrifugation at 16,000 g for 10 min at 4˚C. Protein concentration was determined using Lowry method (Bio-Rad). SDS samples were prepared with Laemmli buffer and heated for 5 min at 95˚C. 10% or 15% SDS electrophoresis gels were used for running and separating protein samples. The proteins were blotted onto nitrocellulose membrane and detected using Ponceau S (Sigma Aldrich, P7170) following which the membrane was destained. Further, nitrocellulose membrane was blocked with 5% milk in 1x TBS-T and probed with the following primary antibodies overnight at 4˚C: MIC26 (#1: Sigma-Aldrich HPA003187, 1:500; #2: Invitrogen PA5-116197, 1:1000; #3: Invitrogen MA5-15493, 1:1000; #4: Pineda home-made, 1:100), MIC27 (Sigma-Aldrich HPA000612, 1:2000), myc-tag (Cell Signaling Technology 2276, 1:1000), ß-Tubulin (Abcam ab6046, 1:2000), Calreticulin (Cell Signaling Technology 2891, 1:1000), ANT2 (Sigma-Aldrich HPA046835, 1:1000) and GFP (Sigma-Aldrich 11814460001, 1:2000). Goat IgG anti-Mouse IgG (Abcam ab97023, 1:10000) and Goat IgG anti-Rabbit IgG (Dianova SBA-4050-05, 1:10000) conjugated to HRP were used as secondary antibodies. The chemiluminescent signals were obtained using Signal Fire ECL reagent (Cell Signaling Technology) and VILBER LOURMAT Fusion SL equipment (Peqlab).

## RNA isolation and quantification

Total RNA was extracted from murine liver tissue using RNeasy Mini Kit (Qiagen) according to the manufacturer's protocol. RNA quality and quantity were assessed using BioSpectrometer (Eppendorf). cDNA synthesis from 1 μg RNA was performed using the GoScript™ Reverse Transcriptase Kit (Promega). Next, quantitative real-time PCR was performed in Rotor Gene 6000 (Corbett Research) using GoTagR qPCR Master Mix (Promega) according to manufacturer's instructions with the following primers:

1. HPRT1 (Housekeeping gene) forward:
   5'- CTGGTGAAAAGGACCTCTCGAAG -3'

2. HPRT1 (Housekeeping gene) reverse:
   5'- CCAGTTTCACTAATGACACAAACG -3'

3. MIC26 forward:
   5'- AGCACCCAAAAAGGACTCGCCT -3'

4. MIC26 reverse:
   5'- GGCTCACAATGATGTCGGAGTTG -3'

5. MIC27 forward:
   5'- GTCTATCTGAAGAATCCTCCGCA -3'

6. MIC27 reverse:
   5'- CAAACAGTCGCTCCTAATGTGGC -3'

$C_t$ values were normalized to housekeeping gene *HPRT1* followed by normalization of $\Delta C_t$ values to average $\Delta C_t$ of db/+ control group.

## Mass spectrometry

Coomassie stained SDS-PAGE cut-outs of the ≈ 47 to 63 kDa band were processed as described before [56] with minor changes: In brief, slices were destained and washed (alternating cycles of 10 mM Ammonium bicarbonate (ABC) and 100 mM ABC/50% Acetonitrile (ACN); 3 repeats). Proteins were reduced and alkylated in-gel (10 mM DTT followed by 55

mM IAA),washed again as before and dried. Digestion was performed for 16 h, 37°C with 0.1 µg trypsin in 100 mM $NH_4HCO_3$. Peptides were extracted twice, using a 1:1 mixture of ACN/0.1% TFA (v/v), combined extracts were dried again and resuspended in 0.1% (v/v) TFA.

Half of each sample was used for LC-MS analysis (Ultimate 3000 rapid separation liquid chromatography system, coupled to a QExactive plus, both Thermo Fisher scientific) using the following parameters: Samples were separated on PepMap (Thermo Fisher scientific) C18 loading and separating columns, using a 2h gradient (going from 4% Buffer B to 40% in 64 minutes, followed by 95%B and 4%B equilibration respectively; Buffer A: 0,1% FA, Buffer B: 84% ACN, 0,1% FA). Data was collected using a resolution of 140k, AGC target of 3e6, maximum injection times of 50ms for MS1 and a scan range of 200-2000m/z, data dependent MS2 spectra were recorded using a resolution of 17,5k, AGC target 1e5, maximum It 50ms, NCE 30 in a TopN [20] experiment using dynamic exclusion for 10 seconds after selection.

Data Processing was performed using MaxQuant 1.6.17.0 [57], using standard parameters if not stated otherwise. Searches were performed against a database containing 79038 human proteins (UP000005640, downloaded 01/2022 from uniport.org). The following modifications were considered: fixed: Carbamidomethylation (C); variable: Oxidation (M); Acetyl (Protein N-term). Potential contaminant IDs were removed and annotation of IDs was performed using Perseus [58], due to the nature of the search, no additional filtering was applied, but results were colour coded, based on the number of peptides assigned to the respective protein IDs in each sample: red: 0 or 1 peptide, green >3 peptides.

### Mouse tissues

Tissues from 12-week-old male control and db/db.BKS mice (#000642, Jackson Laboratories, USA) were used for this study. Animal procedures were approved by the Department for Environment and Consumer Protection of North Rhine-Westphalia, Germany (LANUV; #81–02.04.2019.A321) and the DDZ Institutional Animal Welfare Committee.

## Results

### The presumed 55 kDa glycosylated protein is not decreased in cells depleted for *MIC26* and *MIC27*

Previous studies have either focused on the role of MIC26$_{55kDa}$ glycosylated protein in the context of diabetes and fatty acid metabolism [43,44,47,51,52,54] or MIC26$_{22kDa}$ protein in mitochondrial function [31,46,48,59]. Mitochondrial function and diabetes are intricately connected [60]. Therefore, in order to understand their relationship, we investigated the functional link between MIC26$_{22kDa}$ mitochondrial and MIC26$_{55kDa}$ glycosylated protein. For this, we first focussed on the downregulation of *MIC26* in HEK293 cells (Fig 1A). Western Blots (WBs) consistently revealed a marked reduction of MIC26$_{22kDa}$ protein upon downregulating *MIC26* with three different siRNA, targeting different exons, resulting in negligible amounts of protein levels compared to control siRNA demonstrating the expected sensitivity of MIC26$_{22kDa}$ protein to *MIC26* downregulation. On the contrary, we found no differences in the levels of the MIC26$_{55kDa}$ protein (Fig 1A, uppermost panel) indicating that the MIC26$_{55kDa}$ protein is not affected upon downregulation of *MIC26*. Upon revisiting previous literature, we found two reports showing similar unresponsive nature of the MIC26$_{55kDa}$ glycosylated protein upon downregulation of *MIC26* using multiple siRNA in HeLa cells and shRNA in 143B cells [40,59]. However, this was not further explored. Overall, only the mitochondrial MIC26$_{22kDa}$ protein was responsive while the MIC26$_{55kDa}$ was resistant to *MIC26* downregulation leading us to question the specificity of the MIC26$_{55kDa}$ glycosylated protein.

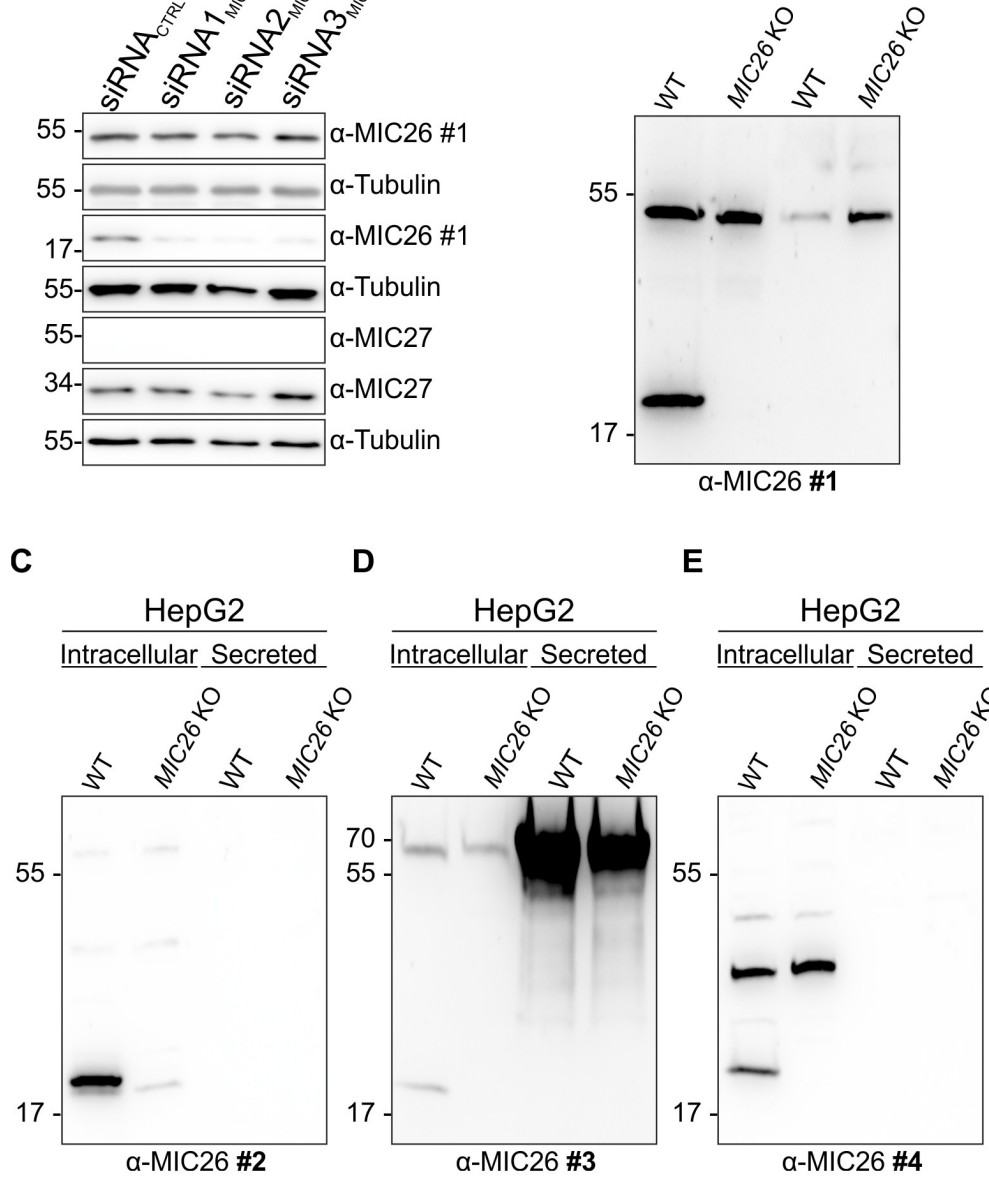

**Fig 1. The MIC26$_{55kDa}$ form is neither sensitive to knockdown nor knockout of *MIC26*.** A) Western blot (WB) analysis of HEK293 cells knocked down for *MIC26* reveal the loss of mitochondrial MIC26$_{22kDa}$ but not MIC26$_{55kDa}$. In order to downregulate *MIC26*, three siRNAs were used. B)–E) WB analysis of cell lysates and secreted proteins, using WT and *MIC26* KO HepG2 cells, were done using four anti-MIC26 antibodies. All four WBs show the presence of intracellular mitochondrial MIC26$_{22kDa}$ form in WT cells but not in *MIC26* KO cells as expected. However, unexpected and inconsistent results were obtained w.r.t the 55 kDa protein. (B) Antibody #1 detects an intra- and extracellular 55 kDa band in WT as well as *MIC26* KOs. (C) Antibody #2 detects only an intracellular protein band around 70 kDa. (D) Antibody #3 identifies an intracellular and secreted protein (in culture medium) at ≈ 70 kDa in both WT and *MIC26* KO HepG2 cells. The considerable increase in ≈ 70 kDa protein correlates to the production and secretion of albumin by human HepG2 cells, which is nonspecifically recognized by this antibody. (E) Antibody #4 is not recognizing a 55 kDa band in WT and *MIC26* KO HepG2 cells.

The previous reports on MIC26$_{55kDa}$ glycosylation had not employed any *MIC26* KO cell line [43,44,47]. One of our studies used *MIC26* and *MIC27* KOs in HAP1 cells but rather focussed on mitochondrial MIC26$_{22kDa}$ and the relationship between mitochondrial ultrastructure and functioning [31]. In order to further explore the unresponsive nature of the MIC26$_{55kDa}$ protein to *MIC26* downregulation, we generated *MIC26* KO in HepG2 cells, using the CRISPR/Cas system, which have appreciable secretory properties and where the MIC26$_{55kDa}$ was described to be secreted [43]. Using WBs, we examined the cell lysates as well as culture media containing the secretory proteins of WT and *MIC26* KO HepG2 cells (Fig 1B) to check for the presence of both the intracellular and secreted MIC26$_{55kDa}$, respectively. Further, in order to corroborate our previous results, we used a set of four anti-MIC26 antibodies available to us. For clarity, we compiled the details of various anti-MIC26 antibodies used in this study and other studies (S1A Fig). The antigens used to elicit antibody response are also shown with different colour-codes from various studies (S1B Fig). All four antibodies consistently detected the human mitochondrial MIC26$_{22kDa}$ in WT HepG2 cell lysates while MIC26$_{22kDa}$ was not observed in *MIC26* KOs as expected (Fig 1B–1E). We observed the putative MIC26$_{55kDa}$ secreted protein in culture media of WT as well as *MIC26* KO HepG2 cells when antibody #1 was used (Fig 1B). Moreover, antibody #1 recognised the presumed intracellular 55 kDa protein (Fig 1B) while a protein band having a molecular weight larger than 55 kDa was detected when antibodies #2 and #3 were used (Fig 1C and 1D). Antibody #2 did not reveal any secreted proteins around 55 kDa while the nearest intracellular protein detected had a molecular weight well above 55 kDa (Fig 1C). Using antibody #4, neither an intracellular nor secreted protein was found in the 55 kDa region (Fig 1E). Thus, the inconsistent detection of proteins of varied molecular weight around 55 kDa, in accordance with no loss of the 55 kDa protein in HepG2 *MIC26* KOs, with different anti-MIC26 antibodies indicates that MIC26$_{55kDa}$ protein is of nonspecific nature and that MIC26 most likely exists in the mitochondria as a 22 kDa protein across many cell types.

In order to check the mitochondrial localisation of MIC26 using all four antibodies, we isolated mitochondria from HEK293 cell lysates and performed immunoblotting (S2 Fig). As expected, we consistently found that MIC26 was only present in the mitochondrial fraction using all antibodies #1–4 (S2A–S2D Fig). ANT2, a mitochondrial protein was only found in the mitochondrial fraction as expected. We also loaded the fraction which contain the endoplasmic reticulum (ER), Golgi and the rest of cytosolic fraction. While calreticulin, an ER protein, was detected in this fraction as expected, mitochondrial MIC26 was not observed showing that mitochondrial MIC26 is not present in the secretory fraction (S2A–S2D Fig).

## A nonspecific 55 kDa band is still recognised in knockouts of *MIC26* and *MIC27* in various human cell lines

In order to ascertain if the 55 kDa protein belongs to either MIC26 and/or MIC27 and to overcome cell-type specific effects, we employed KOs of *MIC26* and *MIC27* in various cell lines. During the course of this study, we generated corresponding single KOs of *MIC26* and *MIC27* in HEK293, HeLa and HepG2 cells using the CRISPR/Cas system. Additionally, we used *MIC26* and *MIC27* KO HAP1 cells along with WT cells already available from our previous study [31]. Altogether, we used KOs of *MIC26* and *MIC27* belonging to four different human cell lines (Fig 2A–2D). Since the 55 kDa MIC26 protein was consistently detected in mammalian cell lysates and cell culture medium using only one antibody (#1) out of four anti-MIC26 antibodies available (Fig 1B–1E), we performed subsequent experiments with WT, *MIC26* and *MIC27* KOs using antibody #1 (Fig 2A–2D). WBs, using lysates from HEK293 (Fig 2A), HepG2 (Fig 2B), HeLa (Fig 2C) and HAP1 (Fig 2D) cells, revealed that the respective MIC26

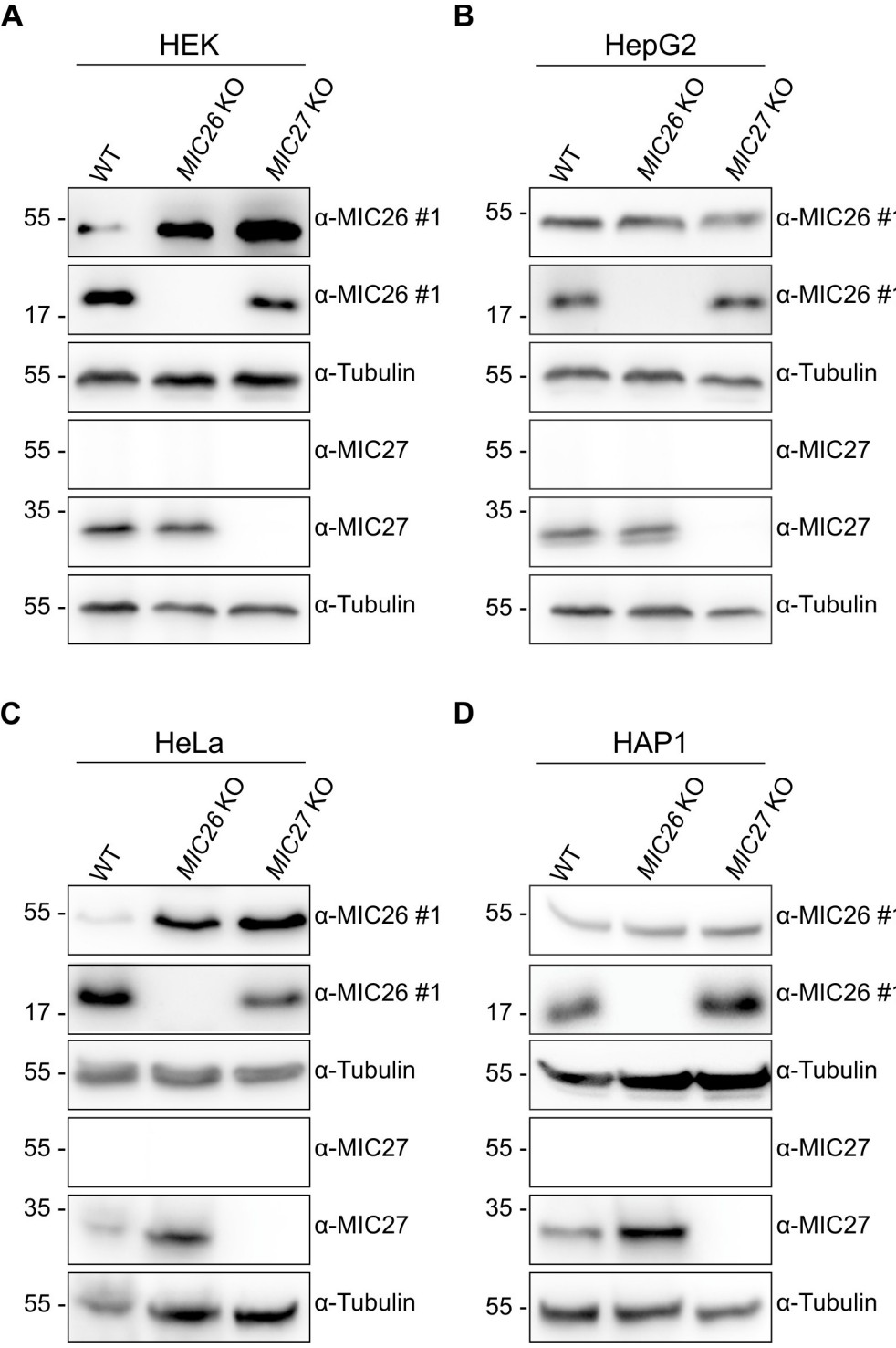

**Fig 2. *MIC26* KOs reveal the nonspecific nature of the 55 kDa band in various human cell lines.** WB analysis of WT, *MIC26* and *MIC27* KO cell lysates, obtained from HEK293, HepG2, HeLa and HAP1 cells are shown. *MIC26* KOs reveal a loss of mitochondrial MIC26$_{22kDa}$ but not the 55 kDa band. There is no protein band detected in WT and *MIC27* KOs at 55 kDa region when anti-MIC27 antibody was used. A) In *MIC26* KOs of HEK293 cells, a loss of MIC26$_{22kDa}$ is observed contrary to an increased amount of 55 kDa protein in WT. B) The KO of *MIC26* in HepG2 cells leads to the loss of the 22kDa form but unchanged expression of the 55 kDa band. C) HeLa cells show similar results to HEK293 cells w.r.t loss of the 22 kDa form and an increased expression of the 55 kDa band. D) In HAP1 cells, the *MIC26* KO leads to a loss of the mitochondrial MIC26$_{22kDa}$ protein contrary to unaltered expression of the 55 kDa protein in WT cells.

and MIC27 proteins were absent when the corresponding genes were knocked out as expected when compared to respective control cells. On the contrary, no loss of 55 kDa protein was observed when using an antibody directed against MIC26 (Fig 2A–2D, topmost panels). Similarly, immunoblotting using an anti-MIC27 antibody does not show a protein in the 55 kDa region in WT cells as well as when cells were knocked out for *MIC27* (Fig 2A–2D and S3A–S3D Fig). In conclusion, the 55 kDa protein belongs neither to MIC26 nor MIC27 confirming that MIC26 and MIC27 only exist as *bona fide* mitochondrial proteins.

## Mutation of various predicted glycosylation sites leads to no change in the level of 55 kDa protein

The experiments involving downregulation (Fig 1A) and KOs of *MIC26* and *MIC27* in multiple human cell lines (Fig 2) using antibody #1 showed that the 55 kDa protein is nonspecific. Further, in order to rule out the possibility that the glycosylation of MIC26 is a possible reason why all four antibodies employed against MIC26 do not recognise the MIC26$_{55kDa}$ protein, we used WT HEK293 cells exogenously expressing MIC26-GFP and probed with an anti-GFP antibody which must recognise GFP of the MIC26$_{55kDa}$-GFP fusion protein. If the MIC26$_{55kDa}$ protein exists, exogenous expression of MIC26-GFP in control HEK293 cells should lead to the formation of 82 kDa protein in addition to the untagged endogenous MIC26$_{55kDa}$ protein. When HEK293 cells expressing MIC26-GFP were probed with an anti-MIC26 (antibody #1) and anti-GFP antibody, a 47 kDa protein (Red asterisks) consistent with the size of mitochondrial MIC26-GFP was observed while no additional band at 82 kDa was detected (Fig 3A and 3B). This provides additional evidence that MIC26 exists only as a mitochondrial protein. In addition, we checked if the presumed MIC26$_{55kDa}$ was sensitive to the disruption of glycosylation pattern. Here, it was predicted *in silico* that three serine amino acid residues of MIC26 could be glycosylated at positions 34, 41 and 50 [46]. Thus, we separately mutated the three serine amino acid residues to alanine to render them incapable of glycosylation. Then, we expressed the theoretically predicted single mutant non-glycosylated versions of MIC26-GFP namely MIC26$^{S34A}$-GFP, MIC26$^{S41A}$-GFP and MIC26$^{S50A}$-GFP in control HEK293 cells and used an anti-MIC26 and anti-GFP antibody to check the size of MIC26-GFP (Fig 3A and 3B). While a 47 kDa protein comprising mitochondrial MIC26-GFP was detected (Fig 3A and 3B) (Red asterisks), a protein corresponding to 82 kDa (55 kDa protein tagged with GFP) was not found (Fig 3A and 3B). When HEK293 cells expressing either MIC26-GFP or the glycosylation defective mutants were further probed with another anti-MIC26 antibody #3, we neither detected a 55 kDa nor a 82 kDa protein (S4 Fig). However, exogenously expressed MIC26-GFP or variants of MIC26-GFP were detected as a 47 kDa protein. It was interesting to note that antibody #3 did not recognise the exogenously expressed MIC26$^{S41A}$-GFP in HEK293 cells showing that serine at position 41 is an important binding site for antibody #3 (S4 Fig). Taken together, we further corroborate that MIC26 only exists as a mitochondrial protein that is not apparently glycosylated.

Since the addition of a large GFP tag could probably disturb the glycosylation pattern of MIC26, we used *MIC26* KOs of HEK293 cells and expressed either an untagged or myc-tagged MIC26 on its C-terminus. WBs of *MIC26* KOs expressing MIC26-myc revealed a mitochondrial myc-specific form, when probed with an antibody against MIC26, that was running slightly above the untagged MIC26 protein (Fig 3C). Expression of MIC26-myc in *MIC26* KOs also revealed a myc-tagged MIC26 when an anti-myc antibody was used. However, we did not observe any increase of the 55 kDa protein while expressing either untagged or myc-tagged MIC26 in *MIC26* KOs. This was also in accordance with a previous report where HeLa cells expressing N-terminal myc-tagged MIC26 were used [46]. Although a mitochondrial

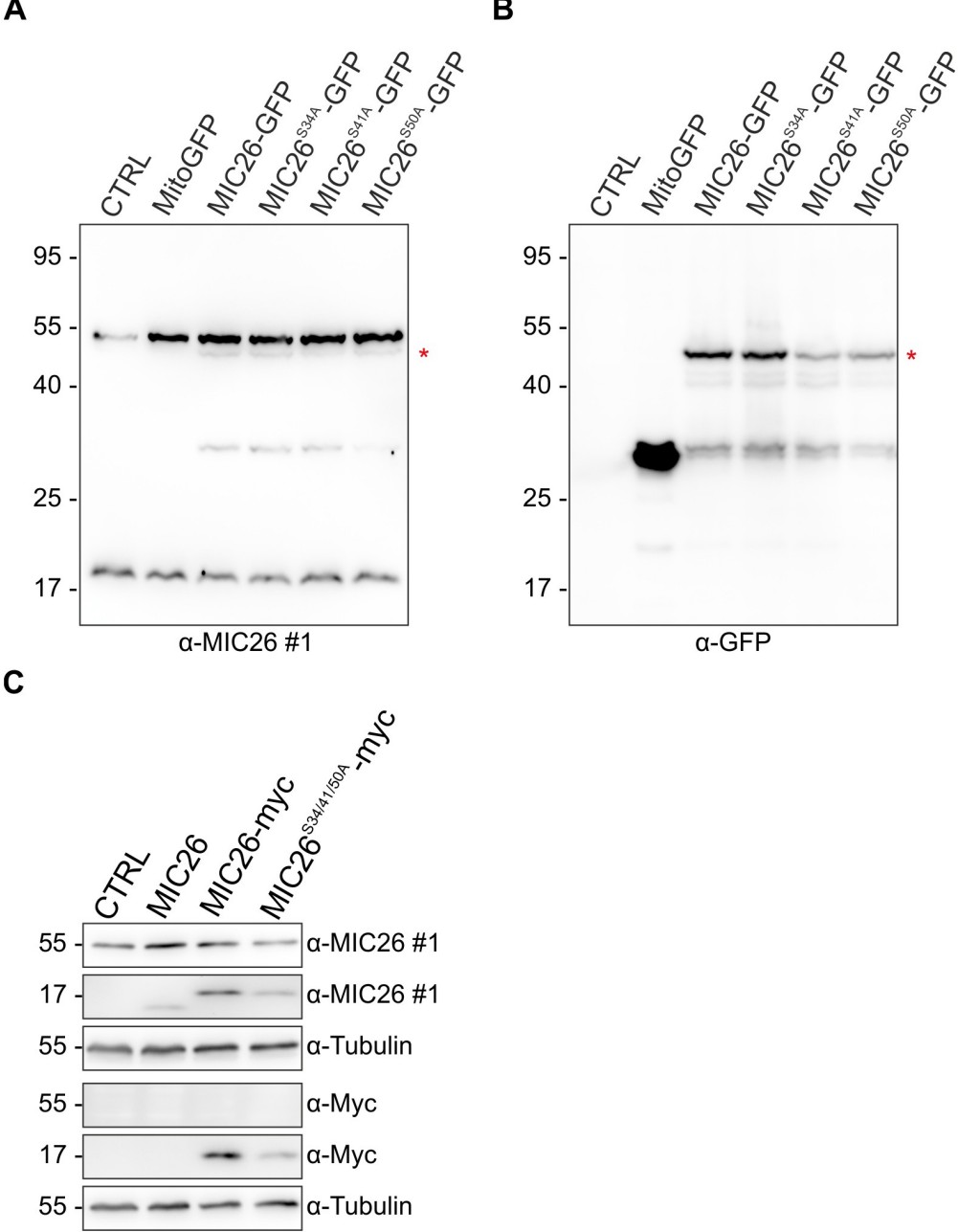

**Fig 3. Exogenous expression of MIC26-GFP or MIC26-myc does not result in the formation of corresponding MIC26$_{55kDa}$-GFP or MIC26$_{55kDa}$-myc protein.** WB analysis of HEK293 cells expressing MIC26-GFP or MIC26-myc along with various theoretically predicted glycosylation defective mutants. A) Overexpression of mitoGFP (control), MIC26-GFP and the single mutants of MIC26-GFP namely MIC26$^{S34A}$-GFP, MIC26$^{S41A}$-GFP and MIC26$^{S50A}$-GFP in HEK WT cells leads to detection of the endogenous MIC26$_{22kDa}$ protein, a MIC26$_{22kDa}$-GFP protein ($\approx$ 50 kDa protein below the 55 kDa protein marked in red asterisks) and the endogenous 55 kDa band when antibody #1 was used. B) Immunoblotting with anti-GFP antibody shows a $\approx$ 50 kDa band detecting MIC26$_{22kDa}$-GFP. However, no band correlating to a MIC26$_{55kDa}$-GFP protein ($\approx$ 80 kDa) could be observed. C) Exogenous expression of MIC26-myc in HEK293 *MIC26* KO leads to the formation of MIC26$_{22kDa}$-myc protein with an expected slight shift in molecular weight compared to the expression of untagged MIC26. However, MIC26$_{55kDa}$-myc protein was neither detected with anti-MIC26 nor anti-myc tag antibody. Expression of the glycosylation defective triple mutant, MIC26$^{S34A/S41A/S50A}$-myc, where the respective serine amino acids were mutated to alanine (S34A, S41A, S50A), showed unaltered 55 kDa protein levels.

MIC26$_{22kDa}$-myc was detected with an anti-myc antibody, no protein was detected at 55 kDa. Further, we generated a C-terminal myc tag plasmid expressing a glycosylation defective triple mutant, S34A/S41A/S50A, in *MIC26* KOs where all three serine amino acids were mutated into alanine. Mutations of all three serine amino acid residues into alanine should lead to an abolished O-linked glycosylation and loss of the 55 kDa protein. However, despite mutating multiple serines into alanine, we neither observed a loss of the 55 kDa protein nor the expression of myc-tag protein in the 55 kDa region (Fig 3C). These observations show that the 55 kDa protein does not originate from MIC26.

## The mitochondrial MIC26 is consistently detected in murine tissues, cells and human cells in contrast to the putative 55 kDa glycosylated version

We have performed *MIC26* downregulation with three different siRNAs in HEK293 cells, probed with four different anti-MIC26 antibodies using control and HepG2 *MIC26* KO cells and further tested *MIC26* KOs in three other human cell lines namely HEK293, HeLa and HAP1 cells (Fig 2), and exogenously expressed MIC26-GFP and MIC26-myc in HEK293 cells (Fig 3 and S4 Fig) which consistently validated that the 55 kDa protein of MIC26 does not exist. In addition, in order to corroborate our observations across species, we compared murine liver lysates from WT C57BL6/J mice with human HEK293 and HeLa cell lysates using multiple antibodies and tested for the presence of a 55 kDa protein. WBs consistently revealed the mitochondrial MIC26$_{22kDa}$ (Fig 4A and 4B, S5A and S5B Fig) reiterating that all four antibodies recognise the correct protein in human cell lines. In the respective WBs, we also loaded lysates of murine liver for comparing the immunoreactivity of anti-MIC26 antibody in human cells and murine tissue. In congruence with human cell line data, antibodies #1 and #2 detected the mitochondrial MIC26$_{22kDa}$ protein in murine liver (Fig 4A & 4B). Hence, antibody #1 and #2 could be used to detect the specific mitochondrial MIC26$_{22kDa}$ protein in human and murine samples. However, one must be cautious in interpreting a band with an apparent molecular weight around 55 kDa using antibody #1 (Fig 4A) as this is not specific to MIC26. Further, we used plasma from WT C57BL6/J mice and found that there is no detection of any protein of any size using anti-MIC26 antibodies #1 and #2 (Fig 4C and 4D) confirming that MIC26 is not secreted into murine plasma. In addition, in order to check whether MIC26$_{22kDa}$ is recognised by both these antibodies in cells having murine origin along with other murine tissues, we used murine lysates of white adipose tissue, muscle, liver and plasma along with C2C12 myoblast cell line and found that antibody #1 and #2 consistently recognise the mitochondrial MIC26$_{22kDa}$ (Fig 4E and 4F).

Further, we compared the WBs of HEK293 and HeLa cell lysates with murine liver lysates using two other antibodies (#3 and #4) against MIC26 (S5A and S5B Fig). As shown before (Fig 1D and 1E), antibodies #3 and #4 recognised the human cell lysates (S5A and S5B Fig). However, antibody #3 did not recognise the murine MIC26$_{22kDa}$ protein (S5A Fig) but recognised nonspecific proteins in murine liver lysates which includes a $\approx$ 25 kDa (Green asterisks) and a 55 kDa protein (S5A Fig). Since the antibody #3 was raised in mouse, the 25 and the 55 kDa proteins probably correspond to the IgG light and heavy chains which were only observed in murine plasma (S5C Fig), liver, muscle and white adipose tissue but not in C2C12 myoblast cell lysate (S5E Fig). We conclude that antibody #3 is not suitable for detecting MIC26 in murine samples. Next, in order to validate whether the 55 kDa protein is recognised in a nonspecific manner in murine samples, we used our custom-made antibody #4 which was designed to immunoreact only to human MIC26$_{22kDa}$ and not murine MIC26$_{22kDa}$. The epitope used to elicit the anti-MIC26 antibody production in rabbit contained a total of 12 amino acids which perfectly match the human MIC26 but not mouse MIC26 as there was a four

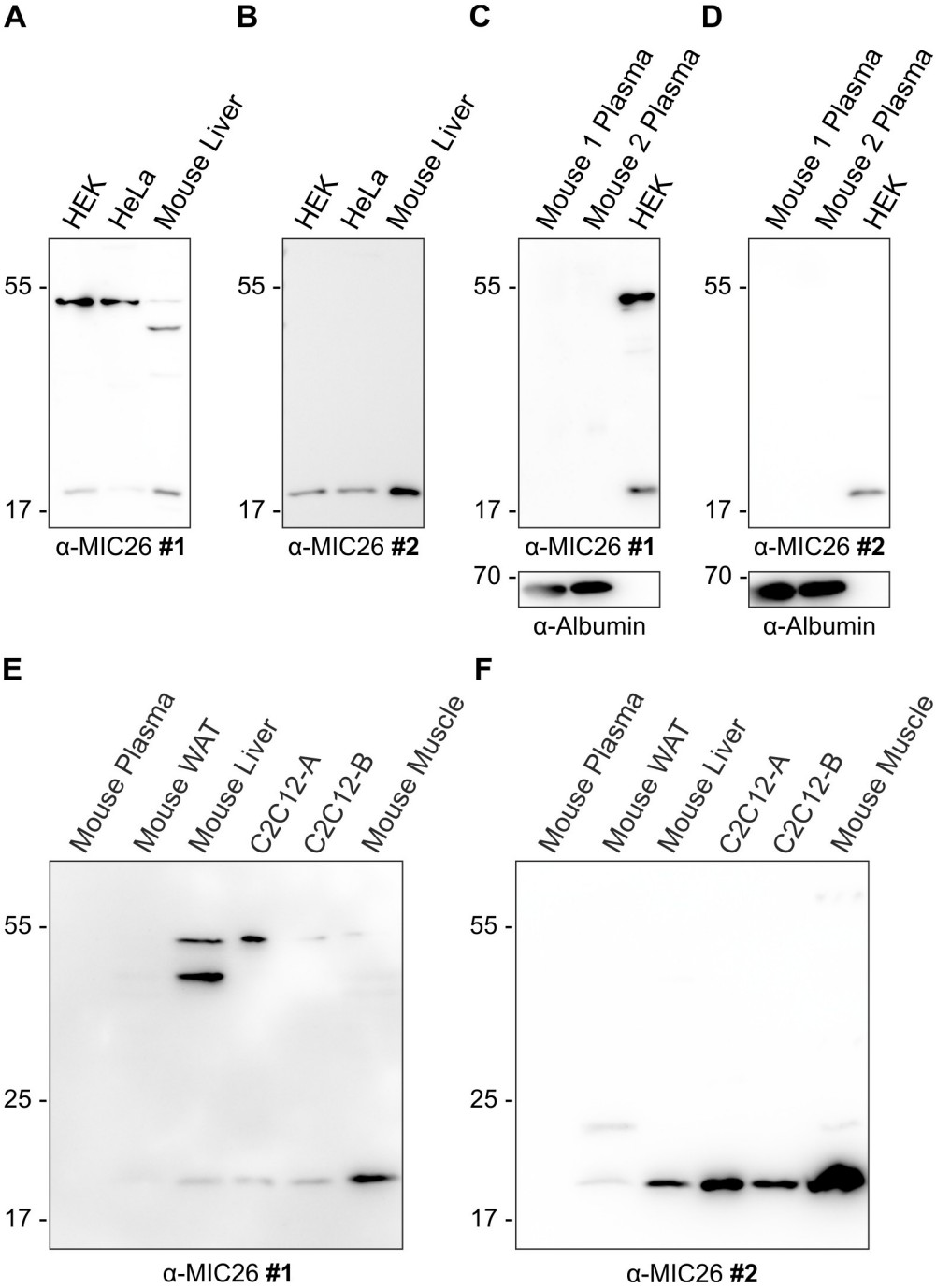

**Fig 4. Unspecific nature of the 55 kDa band is neither cell line- or species-specific.** WB analysis of human WT HEK293 and HeLa cell lysates along with murine liver tissue lysates and plasma comparing two different anti-MIC26 antibodies: A) Antibody #1 detects MIC26$_{22kDa}$ and an unspecific 55 kDa protein in HEK293 and HeLa as well as in murine liver. B) Antibody #2 exclusively detects the mitochondrial MIC26$_{22kDa}$, but no 55 kDa band. C) Antibody #1 does not recognize a mitochondrial MIC26$_{22kDa}$ and 55 kDa band in plasma samples from two mice. D) Antibody #2, like antibody #1, does not recognize any protein in murine plasma samples. E) Antibody #1 detects the MIC26$_{22kDa}$ as well as an unspecific 55 kDa band in C2C12 cell line along with different murine tissue lysates but not plasma. F) Antibody #2 detects MIC26$_{22kDa}$ in mouse cell lines and tissues but not the 55 kDa band. C2C12-A represents C2C12 cells lysed with RIPA buffer and C2C12-B represents C2C12 cells which were mechanically lysed.

amino acid mis-alignment (S1C Fig). Accordingly, on one hand, in murine liver samples, we did not observe a mitochondrial MIC26$_{22kDa}$ band while observing a presumed nonspecific band at various sizes including a 55 kDa band (S5B Fig). On the other hand, in human HEK293 and HeLa cells, antibody #4 showed multiple bands including the MIC26$_{22kDa}$ and a protein at the 55 kDa region (S5B Fig). Therefore, antibody #4 specifically recognised human mitochondrial MIC26$_{22kDa}$ while multiple other nonspecific bands were detected on the WBs in murine liver, HEK293 and HeLa cell lysates including a 55 kDa protein (S5B Fig). Clearly, this inconsistent detection of the 55 kDa protein upon using different antibodies in mammalian HEK293, HeLa cell and murine liver lysates strongly argues further for its nonspecific nature. Thus, while all the four antibodies were recognising the mitochondrial MIC26$_{22kDa}$ version in human cell lysates, only two antibodies (#1 & #2) used in this study are suitable to recognise MIC26$_{22kDa}$ mitochondrial version in murine tissues and cells.

## Mass spectrometry data of 55 kDa protein does not reveal MIC26-specific peptides

The experiments performed in this study conclude that the presumed 55 kDa protein recognised using anti-MIC26 antibodies is nonspecific. Finally, we tested for the presence of MIC26-specific peptides using mass spectrometry in WT, *MIC26* and *MIC27* KO HEK293 cells. For this, we ran the corresponding cell lysates on an SDS gel after which the protein bands at and around the 55 kDa region ($\approx$ 47 to 63 kDa proteins) were excised and the samples were given for mass spectrometry (MS) analysis. We could detect a total of 1273 proteins in the excised gel pieces in all the three samples. In total 278 mitochondrial proteins were detected including e.g. Prohibitin-2, NADH dehydrogenase flavoprotein 1, TIM50, glycerol kinase 2. However, as expected, we did not find MIC26-specific peptides in the MS analysis in any of the three samples analysed which included WT, *MIC26* and *MIC27* KOs. The list of detected peptides is presented in the S1 Data file. Hence, all the experiments presented in this study including the MS results conclusively show that the 55 kDa protein is of nonspecific nature with respect to MIC26.

## The levels of the MIC26$_{22kDa}$ mitochondrial protein are not altered in heart and livers of a db/db diabetic mouse model

*MIC26* mRNA was previously found to be increased in hearts of people with diabetes mellitus [43]. *MIC26* was also elevated in the transcriptome of hearts of dogs fed with high-fat diet [50]. Therefore, we examined whether MIC26 levels were changed in a murine model of obesity-related diabetes. For this, we used db/db.BKS mice which during the first three months develop massive obesity and uncontrolled diabetes due to a mutation in the leptin receptor [61–63]. The lean db/+ mice are normoglycemic and served as controls. We obtained heart and liver tissues from three male diabetic (db/db) and control (db/+) mice at the age of 12 weeks (hence diabetic) [64] and performed WBs using antibody #2 which specifically recognised only the MIC26$_{22kDa}$ mitochondrial band in heart and liver of control and diabetic mice (Fig 5A and 5C). Quantification of the mitochondrial MIC26$_{22kDa}$ protein level (Fig 5B and 5D) revealed no change in heart and liver tissues between control and diabetic mice. Additionally, we checked *Mic26* and *Mic27* transcript levels in liver of db/db.BKS diabetic and control mice and found no differences (Fig 5E). Thus, we conclude that in this standard mouse model for obesity and diabetes, we do not observe differences of MIC26$_{22kDa}$ protein levels in the heart and liver tissues compared to control mice.

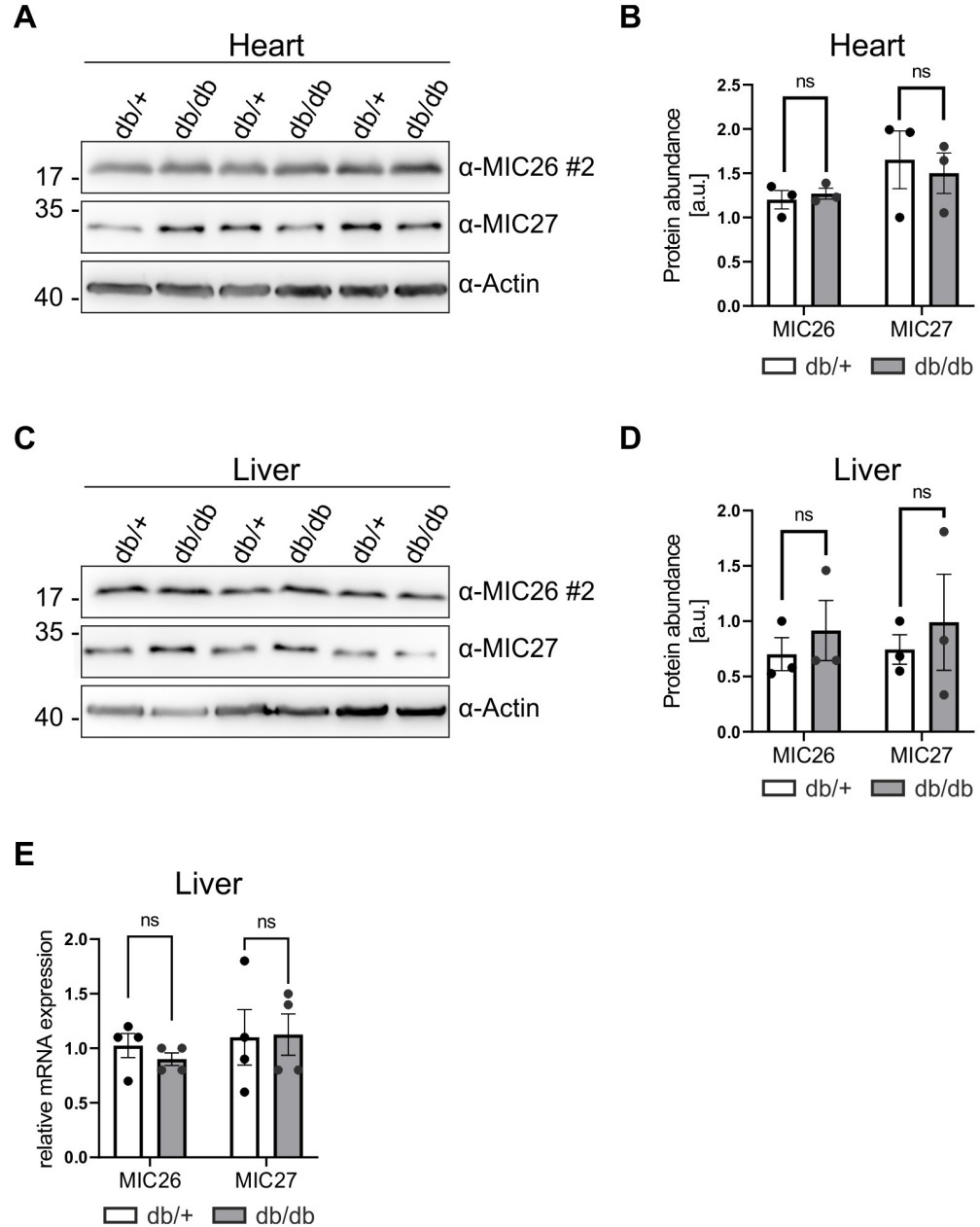

**Fig 5. The diabetic db/db mouse model does not show different levels of the exclusively mitochondrially localized MIC26 and MIC27 proteins in heart and liver.** WB analysis of lysates from heart (A) and liver (C) tissue, along with associated quantification showing protein levels of MIC26 and MIC27 (B and D) from control db/+ and diabetic db/db.BKS mice, reveal no significant differences in amounts of MIC26 and MIC27. E) *Mic26* and *Mic27* transcript levels are not changed in liver of control db/+ and db/db.BKS diabetic mice.

## Discussion

The MIC26 protein was first identified in 2006 where it was detected as a 55 kDa protein in the serum of human, mouse, dogs and in human heart and HepG2 cells despite the size of the recombinant protein being only 22 kDa [43]. Other studies convincingly showed MIC26$_{22kDa}$ was found to be localised to mitochondria which performs important roles in mitochondrial

ultrastructure and function in mammalian cell lines [31,46,48]. In fact, overexpression or downregulation of *MIC26* led to many defects in mitochondrial function [46,59]. During the course of experiments performed for this study, we found that downregulation of *MIC26* using three different siRNAs resulted in a drastic decrease of the mitochondrial $MIC26_{22kDa}$ but showed no effect on the $MIC26_{55kDa}$. In fact, the insensitivity of the $MIC26_{55kDa}$ to the downregulation of the *MIC26* was also observed in other publications but was not focussed upon [40,59]. Further, we employed the *MIC26* as well as *MIC27* KOs in HepG2, HeLa, HEK293 and HAP1 cells. The KO cell lines of *MIC26* did not show a decrease of the presumed 55 kDa protein of MIC26. This is the first report which focussed on the presence of 55 kDa in *MIC26* KO cell lines in combination with multiple validated antibodies. Next, when we expressed exogenous *MIC26-GFP* and *MIC26-myc* in WT and *MIC26* KOs of HEK293 cells respectively and probed with an antibody against MIC26 and GFP or MIC26 and myc, we were not able to detect the corresponding 55 kDa fusion protein of MIC26. However, the mitochondrial $MIC26_{22kDa}$ was consistently observed. A previous study overexpressing *Mic26* in H9C2 rat cardiac myoblast cells led to the formation of a protein which had the same size as recombinant mitochondrial 22 kDa MIC26 protein. Thus, we suppose that the detected protein was mitochondrial $MIC26_{22kDa}$ and not the 55 kDa protein as defined in a previous report [47]. In order to verify if the presumed 55 kDa MIC26 protein was detectable in other species, we studied plasma, tissue and cell lines of murine origin using four antibodies available. Two out of four antibodies are able to detect murine MIC26, while the other two antibodies exclusively showed human specific reactivity. Using an antibody which only showed species reactivity with human samples resulted in no detection of a MIC26 mitochondrial version in murine tissues as expected, while the presumed $MIC26_{55kDa}$ could still be detected. Different to the immunoreactivity of murine tissues, when we used cell lysates from HEK293 and HeLa cells, we consistently detected the $MIC26_{22kDa}$ band but not the $MIC26_{55kDa}$ band. Therefore, experiments involving RNAi, KO cell lines and overexpression of *MIC26* in combination with multiple antibodies demonstrate that the reported 55 kDa protein band does not represent MIC26. We were interested to check the $MIC26_{22kDa}$ in a well-established diabetic mouse model, db/db.BKS, since *MIC26* transcripts were found to be upregulated in the hearts of people with diabetes [43]. However, we were not able to detect differences on protein level in the heart and liver tissue lysates for MIC26 and MIC27. This discrepancy could arise from increased degradation of MIC26 in db/db mice compared to control mice which probably results in increased levels of MIC26 transcripts as a compensatory mechanism.

Previous studies showed that a $MIC26_{55kDa}$ was invariably observed [43,46,47,53,54,65] and several indications supported the wrong interpretation that this form is a glycosylated variant of MIC26. A cleavage of the presumed $MIC26_{55kDa}$ into an apparent 22 kDa form was observed after cABC treatment [43,46]. Similarly, treatment of cells with an inhibitor of glycosylation by using *p*-nitrophenyl-*β*-D-xyloside (PNPX) led to a partial increase of the $MIC26_{22kDa}$ band [43] which, however, was not very efficient. In addition, the secretion of MIC26 into the medium was decreased by using microsomal triglyceride transfer protein (MTP) inhibitors like Naringenin and CP-346086. However, even the secretion could only be inhibited to a maximum of 50%. Therefore, on one hand the authors of this manuscript stated that the $MIC26_{22kDa}$ is a result of cleaving the O-linked glycosylation of $MIC26_{55kDa}$, while on the other hand the inefficient cleavage of $MIC26_{55kDa}$ into $MIC26_{22kDa}$ was explained by $MIC26_{55kDa}$ possessing alternate glycosylation patterns. Further, while it is plausible that a protein could have multiple post-translational modifications including glycosylation, the specificity of the $MIC26_{55kDa}$ as a *bona fide* version of MIC26 protein was not validated through any depletion experiments in previous studies, which we have performed in multiple ways in this study, and which shed doubt over the nature of $MIC26_{55kDa}$ protein. In fact, when a transgenic

mouse overexpressing *Mic26* was used, the transcript levels of *Mic26* were roughly doubled as expected [47]. However, WBs showed a disproportionate increase in the MIC26$_{55kDa}$ band in this study. This could be explained by the fact that the antibody used to detect the 55 kDa band is promiscuously recognising another secreted protein which is affected due to the *MIC26* KOs. It is further possible that the unspecific protein being promiscuously recognised is also sensitive to partial cleavage by cABC which could explain previous data that glycosylation inhibitors like PNPX result in reduced amounts of the assumed 55 kDa glycosylated form of MIC26 or the apparent cleavage of the nonspecific 55 kDa protein after cABC treatment [43,46]. Still, we asked whether the 55 kDa protein of MIC26 really exists and is being secreted by checking the previous human secretome studies. 2641 proteins were predicted to be comprising the revised whole human secretome [65]. MIC26 despite being predicted to be secreted in the human blood was not empirically found in mass spectrometry analysis, antibody-based immunoassays and proximity extension assays [65]. Consistently, MIC26 was not found as a protein secreted by HepG2 cells [66,67]. Previously, it was found that oleic acid treatment of HepG2 cells increased the protein levels of the 55 kDa of MIC26 [52,54]. However, the secretome profile of HepG2 cells did not include MIC26 among the list of upregulated or downregulated proteins when cells were treated with oleic acid compared to untreated cells [66]. Accordingly, when we eluted proteins in the range of 47 to 63 kDa and analysed a list of 1273 proteins detected by MS, we were not able to detect MIC26-specific peptides in control cells as well as *MIC27* KOs. All the above observations provide evidence that the 55 kDa protein originally assigned as MIC26 is of nonspecific nature.

Various monoclonal and polyclonal antibodies respond differently with respect to the recognition of the presumed 55 kDa protein band of MIC26 in various studies (commercial or custom-made). In one of the initial studies, a mixture of two epitopes were used to generate an anti-MIC26 antibody where one epitope covered the N-terminus from amino acids 22 to 36 whereas another epitope covered the C-terminus covering amino acids 184–198. This antibody strongly detected the 55 kDa protein. Antibody #1 was made using an epitope of 66 amino acids covering C-terminus region from 133–198 of MIC26 whereas antibody #2 was made where the epitopes covered amino acids 9 to 38. While antibody #1 recognises a 55 kDa band, antibody #2 does not recognise it. Therefore, taken together it is possible that the nonspecificity of the 55 kDa form arises from the C-terminal region as antibodies made from exposure to N-terminal epitope do not recognise the 55 kDa region. Overall, this study demonstrates the importance of antibody validation using respective KO cell lines and gives valuable hints how to study the role of MIC26 in the future.

## Supporting information

**S1 Checklist. The ARRIVE guidelines 2.0: Author checklist.**
(PDF)

**S1 Fig. Overview and alignment of various anti-MIC26 antibodies.** A) Summary of seven anti-MIC26 antibodies used in this study (#1-#4) and other (#1, #3, #5-#7) publications are described. The figure provides 1) the source of the antibody, 2) the host and region of the peptide used for antibody generation (amino acid numbers: aa), 3) the related color-code to (B), 4) publications describing the usage of the respective antibody, 5) if studies were performed regarding the 55 kDa form of MIC26 and if yes 6) which cell lines or tissues have been investigated. N/A is Not Applicable. B) Alignment of the human and mouse MIC26 amino acid sequence shows an identity of 83%. The epitopes used to generate different anti-MIC26 antibodies are highlighted by color-code and further described in the figure above. C) Alignment of the human and mouse MIC26 amino acid sequences, from position 61 to 110, with 12 amino-

acid epitope used to generate antibody #4 (Home-made, Pineda) which shows proper alignment of antibody #4 epitope to human MIC26, but not mouse MIC26 where four mismatches were detected. Red color shows eight amino acids which are matching out of 12 amino acids.
(TIF)

**S2 Fig. MIC26 is exclusively present in mitochondrial fractions.** A) WBs showing the total cellular fraction, mitochondrial fraction as well as a fraction containing combined ER, Golgi and remaining cytosol, when anti-MIC26 antibody #1 was used. Antibodies against ANT2 and calreticulin served as markers for the mitochondrial as well as the combined ER, Golgi and cytosolic fraction respectively. B-D) WBs showing different fractions mentioned above where anti-MIC26 antibodies #2–4 were used.
(TIF)

**S3 Fig. A 55 kDa band is not detected by anti-MIC27 antibody.** A-D) WB analysis of cell lysates from HEK293 (A), HepG2 (B), HeLa (C) and HAP1 (D) cells using anti-MIC27 antibody in WT, *MIC26* KO and *MIC27* KOs. Whole WBs are shown for clarity.
(TIF)

**S4 Fig. Exogenous MIC26-GFP expression does not generate MIC26$_{55kDa}$-GFP.** Antibody #3 detects the endogenous MIC26$_{22kDa}$ protein as well as MIC26$_{22kDa}$-GFP. However, no MIC26$_{22kDa}$-GFP protein was detected for the S41A mutant, leading to the assumption, that the antibody #3 has a strong binding affinity for serine in position 41. Furthermore, in accordance with anti-MIC26 antibody #1 and anti-GFP antibody (Fig 3), a $\approx$ 80 kDa MIC26$_{55kDa}$-GFP protein was not recognized.
(TIF)

**S5 Fig. MIC26 antibody #3 and #4 recognize human but not murine MIC26$_{22kDa}$.** A) Antibody #3 shows immunoreactivity against MIC26$_{22kDa}$ in human cell lines but not murine liver tissue lysates, providing evidence regarding the unspecific nature of the detected 25 kDa and 55 kDa proteins in liver lysates. Additionally, antibody #3 shows an unspecific binding of $\approx$ 70 kDa protein in human cell lines. B) Antibody #4, comparable to antibody #3, shows immunoreactivity against MIC26$_{22kDa}$ in human cell lines but not murine liver tissue lysate revealing the unspecific nature of several additional bands detected in murine liver. C) Antibody #3 nonspecifically recognizes a 25 kDa and a 55 kDa protein in two different murine plasma samples, probably derived from light and heavy chain of IgG. D) Antibody #4 detects an unspecific band at approximately 70 kDa in murine plasma samples. E) Antibody #3 is not able to recognize MIC26$_{22kDa}$ protein in murine samples. However, it recognizes a 25 kDa and a 55 kDa band only in murine tissue samples but not in mouse cell lines indicating a nonspecific detection of the light and heavy chain of IgG. C2C12-A represents C2C12 cells lysed with RIPA buffer and C2C12-B represents C2C12 cells which were lysed mechanically.
(TIF)

**S1 Data. Excel sheet of mass spectrometry analysis.** The excel sheet depicts all identified peptides from mass spectrometry analysis of HEK293 WT, *MIC26* and *MIC27* KO cell lines, which were isolated from an SDS gel, having a molecular weight of $\approx$ 47–63 kDa. Sheet 1 contains all detected proteins, while sheet 2 is filtered only for mitochondrial proteins identified from Gene Ontology (GO) cellular compartments.
(XLSX)

**S1 Raw images. Uncropped blots of all blots shown in the manuscript.**
(PDF)

## Acknowledgments

We thank Tanja Portugall for technical assistance in constructing the plasmids used in this study. Mass spectrometry experiments were performed at the proteomics facility, BMFZ, HHU, Düsseldorf. FACS cell sorting was performed at ITZ, UKD, HHU, Düsseldorf by Katharina Raba.

## Author Contributions

**Conceptualization:** Arun Kumar Kondadi.

**Formal analysis:** Melissa Lubeck, Ritam Naha, Marc D. Driessen.

**Funding acquisition:** Ruchika Anand, Andreas S. Reichert, Arun Kumar Kondadi.

**Investigation:** Melissa Lubeck, Nick H. Derkum, Marc D. Driessen, Ruchika Anand.

**Methodology:** Melissa Lubeck, Arun Kumar Kondadi.

**Project administration:** Arun Kumar Kondadi.

**Resources:** Rebecca Strohm, Bengt-Frederik Belgardt, Michael Roden, Kai Stühler, Andreas S. Reichert.

**Supervision:** Andreas S. Reichert, Arun Kumar Kondadi.

**Validation:** Melissa Lubeck.

**Visualization:** Melissa Lubeck, Ritam Naha.

**Writing – original draft:** Arun Kumar Kondadi.

**Writing – review & editing:** Melissa Lubeck, Bengt-Frederik Belgardt, Michael Roden, Ruchika Anand, Andreas S. Reichert, Arun Kumar Kondadi.

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
