## [Decision Letter · Decision Letter 0]

14 Feb 2023

PONE-D-22-33726MIC26 and MIC27 are bona fide subunits of the MICOS complex in mitochondria and do not exist as glycosylated apolipoproteinsPLOS ONE

Dear Dr. Kondadi,

Thank you for submitting your manuscript to PLOS ONE. After careful consideration, we feel that it has merit but does not fully meet PLOS ONE’s publication criteria as it currently stands. Therefore, we invite you to submit a revised version of the manuscript that addresses the points raised during the review process. 

The reviewers recognized the importance of your study, but they have minor points that need to be clarified.

Please revise the manuscript according to the reviewer' comments.

We look forward to receiving your revised manuscript.

Kind regards,

Benedetta Ruzzenente

Academic Editor

PLOS ONE

Journal Requirements:

2. As part of your revision, please complete and submit a copy of the Full ARRIVE 2.0 Guidelines checklist, a document that aims to improve experimental reporting and reproducibility of animal studies for purposes of post-publication data analysis and reproducibility: https://arriveguidelines.org/sites/arrive/files/documents/Author%20Checklist%20-%20Full.pdf Please include your completed checklist as a Supporting Information file. Note that if your paper is accepted for publication, this checklist will be published as part of your article.

In your cover letter, please note whether your blot/gel image data are in Supporting Information or posted at a public data repository, provide the repository URL if relevant, and provide specific details as to which raw blot/gel images, if any, are not available. Email us at plosone@plos.org if you have any questions

Reviewers' comments:

Reviewer's Responses to Questions

**Comments to the Author**

1. Is the manuscript technically sound, and do the data support the conclusions?

Reviewer #1: Yes

Reviewer #2: Yes

Reviewer #3: Yes

2. Has the statistical analysis been performed appropriately and rigorously? 

Reviewer #1: N/A

Reviewer #2: Yes

Reviewer #3: Yes

3. Have the authors made all data underlying the findings in their manuscript fully available?

Reviewer #1: Yes

Reviewer #2: Yes

Reviewer #3: Yes

4. Is the manuscript presented in an intelligible fashion and written in standard English?

Reviewer #1: Yes

Reviewer #2: Yes

Reviewer #3: Yes

5. Review Comments to the Author

Reviewer #1: In their manuscript submitted to PLoS ONE, Lubeck et al. investigate the 55 kDa form of the MICOS component MIC26 that was initially reported to be a glycosylated, secreted version of the 22 kDa protein. This protein was described to be present in lipoproteins, leading to the categorization of MIC26 as apolipoprotein O (APOO) and, subsequently, of the related MICOS subunit MIC27 as apolipoprotein O-like (APOOL). Biochemical studies of the two proteins have focused on their function in mitochondria as part of MICOS, and the function of the 55 kDa protein has remained unclear.

Surprisingly, the authors demonstrate conclusively that this mysterious protein is not a derivate of MIC26 or of MIC27, and instead represents an unrelated protein that is crossreactive with certain anti-MIC26 antibodies. Moreover, they report that there is no specific secreted species of MIC26 in cell culture supernatant or in mouse plasma, which agrees with analyses of secretomes. These findings suggest that MIC26 is localized only within the cell, in line with its well established mitochondrial function.

This study is an important correction to the body of MICOS literature and will be crucial to ongoing research focusing on the 55 kDa species, an unidentified protein with possibly important functions.

Some points that should be addressed:

1. The mouse plasma Western blots in Fig. 4C,D are empty. As positive controls, authentic MIC26 from cell extracts as well as an established plasma protein need to be included on the same gel.

2. The authors discuss the inefficient deglycosylation of the 55 kDa protein in the original study (Lamant et al., 2006), but not that they themselves have reported quite efficient conversion of the 55 kDa band into what they thought was deglycosylated MIC26 (Koob et al., 2015, BBA). How do the authors explain this? Presumably the lower MW band that appears after treatment is the unglycosylated version of the 55 kDa protein = not MIC26 but still recognized by the MIC26 antiserum. The authors should provide evidence in an experiment analogous to their published one, that the deglycosylated protein is not recognized by more specific antisera, and that the assay yields the same results in MIC26 deletion cells.

3. Since most of the original criteria for the inclusion of MIC26 into the apolipoprotein family have been disproven in this study, the authors should discuss whether the last claim from the original study, sequence similarity to apolipoproteins, has any merit and may be related to lipid binding by MIC26 or MIC27. This issue, as well as the question whether or not their classification as APOO and APOOL should be rejected, is of continued relevance to the field.

4. It would be helpful if the authors performed a subcellular fractionation to corroborate with different antibodies that MIC26 does not enter the secretory pathway.

Reviewer #2: This paper analyzes the 55 kDa version of the Mic26 protein that has been described to be a glycosylated Mic26 in some publications. Using knockout and different antibodies, as well as mutagenesis of glycosylation sites, the expression of the GFP-tagged protein, and mass spectrometry, the authors show that the observed 55 kDa band is not the glysosylated Mic26, but most likely an antibody-specific cross-reactive band.

In my opinion, the paper successfully proves the point and is an important contribution to the field, because it shows that Mic26 exists only as mitochondria-associated protein. The question remains, however, what protein was identified in the serum using ApoO antibody (Lamant et al, JBC, 2006) and why this antibody did not show the presence of the 22 kDa form of ApoO in the cell lysate. However, it is important to present convincing data that strongly disprove that the observed 55 kDa band was ApoO and open the previously published results to discussion. For that reason, I support the publication of this study.

I have only minor comments to the paper:

- The authors show that there is no increase in protein levels of Mic26/ApoO in their model of diabetic mouse. Because these data partially contradict the previously published information about the upregulation of the Mic26 transcription in diabetes, the authors should check the transcription of Mic26 in their mouse model and show this data in addition to the western blot data.

- Would the authors venture to propose what the origin of the 55 kDa band is? An immunoprecipitation with the antibody that recognizes the 55 kDa band might give an answer to this and explain the observations from the Lamant et al paper.

Text correction: Line 572, N-Terminus should be N-terminus.

Reviewer #3: The aim of the manuscript by Lubeck at al. is to resolve once and for all the false notion that the apoplipoprotein subunit of human MICOS Mic26 exists as two isoforms: one bona fide subunit of the MICOS complex as a 22 kDa form and a secreted, glycosylated 55 kDa form. This very thorough study uses a battery of KO human cell lines, expression of tagged forms, putative glycosylation mutants, antibodies and RNAi knockdowns to support their conclusion that the 55 kDa form detected using one of the Mic26 antibodies (antibody #1) is not derived from the 22 kDa form. Furthermore, they eliminate the possibility that this 55 kDa band can arise from the Mic27 paralog. Finally, they show that levels of Mic26 protein is not affected in the diabetes mouse model, thanks to their efforts to show that antibodies #1 and #2 raised against the human ortholog crossreacts with the mouse; this result contradicts transcriptomic data claiming Mic26 mRNA is affected in these mice. The manuscript is well written and presents novel findings that are appropriate for publication in PLoS One. The presented data overall supports the conclusions made in the paper. Thus, I recommend the manuscript be accepted after minor revision. I list below some minor points that I think should be addressed by the authors to finalize the manuscript.

1) The 55 kDa region of the Western blot probed with the Mic27 antibody is shown in Figures 1 and 2. I am sure the authors are showing this region, but it would be more convincing if they also show at least an exemplar whole blot image, as they have done for the various Mic26 antibodies.

2) I do not find the Mic27 RNAi silencing to be very convincing in Figure 1. The band appears to be at the same level as the control on the very left. No quantification of Westerns from this silencing are shown. Given the authors later utilize the Mic27 KO cell lines to convincingly demonstrate that the 55 kDa band is not derived from Mic27 either, the RNAi-silencing of Mic27 does not seem so relevant here. I would suggest the authors remove this experiment from the manuscript, keeping the robust Mic26 RNAi.

3) Regarding the experiments to remove the serine residues that represent putative O-linked glycolylsation sites. As far as I know, this is not the only amino acid that can undergo this modification. Furthermore, since there is no evidence that the 22 kDa form is glycosylated, the authors should make clear these sites are theoretical and also other amino acids with exposed hydroxyl groups in their side chains can also be I n theory glycosylated. But once again, this is more for transparency. The authors have shown convincingly that the 55 kDa glycosylated protein is not derived from either Mic26 or Mic27.

4) On lines 26-27, the authors state that MICOS is a well-conserved complex. I disagree with the phrasing. Mic60 and Mic10 are well-conserved subunits and the role in making contact sites appears to be conserved in land plants (Michaud et al., 2016 Curr. Biol. 26:627-639), trypanosomes (Kaurov et al., 2018 Curr. Biol. 28:3393-3407) and alphaproteobacteria (Munoz-Gomez et al., bioRxiv, 2022.06. 14.496148). Furthermore, the trypanosome MICOS subunit composition is very different from yeast and human MICOS. This is consient with most of the other MICOS subunits from animals/yeast not distributed in the eukaryotic tree of life in contast to Mic10 and Mic60. This statement should thus be modified to reflect how MICOS as adopted different subunit compositions.

5) On line 236, please cite the already cited Munoz-Gomez et al. (2015) and Huynen et al. (2016) articles, as they show by molecular phyogenetics that human Mic26 and Mic27 are paralogs.

6) Line 390, replace ‘apparently’ with ‘putative’ as the latter seems convey the meaning better.

7) Line 498, delete ‘emphatically’ or replace with another word like ‘convincingly’.

6. PLOS authors have the option to publish the peer review history of their article (what does this mean?). If published, this will include your full peer review and any attached files.

Reviewer #1: No

Reviewer #2: No

Reviewer #3: No

---

## [Author Response · Author response to Decision Letter 0]

19 Apr 2023

28th March 2023, Lubeck et al 2023, PLOS One 

Review Comments to the Author

Reviewer #1: In their manuscript submitted to PLoS ONE, Lubeck et al. investigate the 55 kDa form of the MICOS component MIC26 that was initially reported to be a glycosylated, secreted version of the 22 kDa protein. This protein was described to be present in lipoproteins, leading to the categorization of MIC26 as apolipoprotein O (APOO) and, subsequently, of the related MICOS subunit MIC27 as apolipoprotein O-like (APOOL). Biochemical studies of the two proteins have focused on their function in mitochondria as part of MICOS, and the function of the 55 kDa protein has remained unclear.

Surprisingly, the authors demonstrate conclusively that this mysterious protein is not a derivate of MIC26 or of MIC27, and instead represents an unrelated protein that is crossreactive with certain anti-MIC26 antibodies. Moreover, they report that there is no specific secreted species of MIC26 in cell culture supernatant or in mouse plasma, which agrees with analyses of secretomes. These findings suggest that MIC26 is localized only within the cell, in line with its well-established mitochondrial function.

This study is an important correction to the body of MICOS literature and will be crucial to ongoing research focusing on the 55 kDa species, an unidentified protein with possibly important functions.

Some points that should be addressed:

1. The mouse plasma Western blots in Fig. 4C, D are empty. As positive controls, authentic MIC26 from cell extracts as well as an established plasma protein need to be included on the same gel.

Response: Thank you for the constructive comment. We have now updated the WBs in Fig. 4C and 4D by using HEK293 cell lysates, apart from murine plasma samples, which were immunoblotted using anti-MIC26 antibodies #1 and #2. We used an antibody against albumin as a positive control in WBs using murine plasma which were added during revision. 

In addition, we also updated the S5 Fig.C and S5 Fig.D (previously S3 Fig.C & D), where we performed WBs (using two additional antibodies) using anti-MIC26 #3 and #4 antibodies not only in murine plasma but also in HEK293 lysates as in Fig.4. Anti-albumin antibody was used a positive control for murine plasma.

2. The authors discuss the inefficient deglycosylation of the 55 kDa protein in the original study (Lamant et al., 2006), but not that they themselves have reported quite efficient conversion of the 55 kDa band into what they thought was deglycosylated MIC26 (Koob et al., 2015, BBA). How do the authors explain this? Presumably the lower MW band that appears after treatment is the unglycosylated version of the 55 kDa protein = not MIC26 but still recognized by the MIC26 antiserum. The authors should provide evidence in an experiment analogous to their published one, that the deglycosylated protein is not recognized by more specific antisera, and that the assay yields the same results in MIC26 deletion cells.

Response: Thank you for the suggestion. We have performed experiments with WT and MIC26 KOs in HEK293 cells where the cell lysates were treated with or without chondroitinase ABC (cABC) enzyme and probed with anti-MIC26 antibody #1. We observe a slight reduction in the 55 kDa band, while we do not observe an accumulation in the 22 kDa region for WT as well as MIC26 KOs upon cABC treatment. It suggests that the accumulation (shown in previous manuscript) is only found probably under certain selective conditions. In fact, even after repetitive experiments, we were not able to observe the result which was shown before. Therefore, our observations in previous manuscript could be explained because of nonspecific proteolytic cleavage under specific conditions. 

As a positive control for proper cABC enzymatic digestion upon our treatment, we probed the same cell lysates with an anti-Chondroitin-4-Sulfate antibody (Sigma Aldrich, MAB2030, 1:1000). A clear accumulation of released chondroitin-4-sulfate proteoglycans can be observed in WT and MIC26 KO cells treated with cABC (shown in asterisks). 

(Fig included in original PDF Rebuttal upload)

3. Since most of the original criteria for the inclusion of MIC26 into the apolipoprotein family have been disproven in this study, the authors should discuss whether the last claim from the original study, sequence similarity to apolipoproteins, has any merit and may be related to lipid binding by MIC26 or MIC27. This issue, as well as the question whether or not their classification as APOO and APOOL should be rejected, is of continued relevance to the field.

Response: According to our observations in this manuscript, we have only shown that the 55 kDa protein is nonspecifically recognised by certain anti-MIC26 antibodies. Therefore, MIC26 does not exist as a modified 55 kDa protein. MIC26 (1) and MIC27 (2) sequence analysis have shown that they possess apolipoprotein family domains. Accordingly, MIC27 has been shown to bind to cardiolipin species. Previously, we have also shown that double knockouts of MIC26 and MIC27 led to reduced cardiolipin levels (3). Therefore, while we argue that the glycosylated version of MIC26 does not exist, we think that MIC26 and MIC27 are still apolipoproteins whose function is not completely understood in the mitochondria. Another human apolipoprotein, ApoE is also known to localise to mitochondria (apart from other organelles) and plays an important role in cholesterol metabolism in addition to its secretion in plasma (4). Altogether, MIC26 and MIC27 are mitochondrial apolipoproteins, although MIC26 does not exist as a secreted glycosylated protein. 

4. It would be helpful if the authors performed a subcellular fractionation to corroborate with different antibodies that MIC26 does not enter the secretory pathway.

Response: This is a very helpful suggestion from the reviewer. We have performed subcellular fractionation assays, where mitochondrial fraction was separated from the combined ER/Golgi/cytosolic fraction, using HEK293 cell lysates. Western blotting using anti-ANT2 antibody showed immunopositivity in total and mitochondrial fractions as expected while calreticulin was found in the ER/Golgi/cytosol fraction as expected. Using all four anti-MIC26 antibodies, we found that mitochondrial MIC26 was exclusively present in total and mitochondrial fractions (and not in ER/Golgi/cytosolic fraction) corroborating that MIC26 does not enter the secretory pathway. The new figure is included in the revised version as S2 Figs. A-D. 

Reviewer #2: This paper analyzes the 55 kDa version of the Mic26 protein that has been described to be a glycosylated Mic26 in some publications. Using knockout and different antibodies, as well as mutagenesis of glycosylation sites, the expression of the GFP-tagged protein, and mass spectrometry, the authors show that the observed 55 kDa band is not the glysosylated Mic26, but most likely an antibody-specific cross-reactive band.

In my opinion, the paper successfully proves the point and is an important contribution to the field, because it shows that Mic26 exists only as mitochondria-associated protein. The question remains, however, what protein was identified in the serum using ApoO antibody (Lamant et al, JBC, 2006) and why this antibody did not show the presence of the 22 kDa form of ApoO in the cell lysate. However, it is important to present convincing data that strongly disprove that the observed 55 kDa band was ApoO and open the previously published results to discussion. For that reason, I support the publication of this study.

I have only minor comments to the paper:

- The authors show that there is no increase in protein levels of Mic26/ApoO in their model of diabetic mouse. Because these data partially contradict the previously published information about the upregulation of the Mic26 transcription in diabetes, the authors should check the transcription of Mic26 in their mouse model and show this data in addition to the western blot data.

Response: We thank the reviewer for this constructive criticism. We have used liver tissues from db/db.BKS diabetic mice model and compared them with db/+ control mice for checking the levels of MIC26 transcripts. Our data in Fig. 5E shows that MIC26 as well as MIC27 transcripts are similar in diabetic mice compared to control mice 

In addition to this, we checked the transcriptomics data from Huang et al., 2022 (5), where they performed transcriptome sequencing from heart tissue of BKS wildtype control and db/db.BKS mouse. Briefly, we retrieved the “geneCounts” and “geneFPKM” files from NCBI GEO database (GSE161931) and performed DESeq2 analysis on the count file. We did not observe any significant difference in Fragments Per Kilobase of transcript per Million mapped reads (FPKM) of MIC26 and MIC27 between BKS wildtype control and db/db.BKS mouse. DESeq2 analysis revealed a minor but not significant decrease in log2 Fold Change of MIC26 and a minor increase in MIC27. Considering the transcriptomics data, we did not notice any considerable difference in mRNA expression of MIC26 and MIC27 in db/db.BKS mouse as compared to db/+ wildtype control strain.

(Fig included in original PDF Rebuttal upload)

Would the authors venture to propose what the origin of the 55 kDa band is? An immunoprecipitation with the antibody that recognizes the 55 kDa band might give an answer to this and explain the observations from the Lamant et al paper.

Response: Thank you for the comment. Initially, we were of a similar opinion as the reviewer #2. However, we think that various MIC26 antibodies recognise different proteins which is clear upon careful observation of the blots in Fig 1B-D. In Fig. 1B, the detected intracellular as well as secreted protein in WT and MIC26 KO HepG2 cells runs slightly below 55 kDa marker while the proteins recognised using antibodies #2 (Fig. 1C) and #3 (Fig.1D) run above 55 kDa marker. Therefore, antibody #1 is not recognising the same as antibody #2 and #3. Further, it is not possible that antibodies # 2 and # 3 are recognising the same protein as antibody #2 is not recognising any secreted protein while antibody #3 is recognising secreted proteins above 55 kDa marker band. Thus, due to our observations that multiple proteins are recognised with various antibodies, we were not encouraged to investigate the nature of the 55 kDa protein further. We also think that investigating the nature of these protein/s is beyond the scope of this manuscript.

Text correction: Line 572, N-Terminus should be N-terminus.

Response: The text has been changed accordingly in the new line 619.

Reviewer #3: The aim of the manuscript by Lubeck at al. is to resolve once and for all the false notion that the apoplipoprotein subunit of human MICOS Mic26 exists as two isoforms: one bona fide subunit of the MICOS complex as a 22 kDa form and a secreted, glycosylated 55 kDa form. This very thorough study uses a battery of KO human cell lines, expression of tagged forms, putative glycosylation mutants, antibodies and RNAi knockdowns to support their conclusion that the 55 kDa form detected using one of the Mic26 antibodies (antibody #1) is not derived from the 22 kDa form. Furthermore, they eliminate the possibility that this 55 kDa band can arise from the Mic27 paralog. Finally, they show that levels of Mic26 protein is not affected in the diabetes mouse model, thanks to their efforts to show that antibodies #1 and #2 raised against the human ortholog crossreacts with the mouse; this result contradicts transcriptomic data claiming Mic26 mRNA is affected in these mice. The manuscript is well written and presents novel findings that are appropriate for publication in PLoS One. The presented data overall supports the conclusions made in the paper. Thus, I recommend the manuscript be accepted after minor revision. I list below some minor points that I think should be addressed by the authors to finalize the manuscript.

1) The 55 kDa region of the Western blot probed with the Mic27 antibody is shown in Figures 1 and 2. I am sure the authors are showing this region, but it would be more convincing if they also show at least an exemplar whole blot image, as they have done for the various Mic26 antibodies.

Response: Thank you for the comment. We have now included whole Western blots of WT, MIC26 and MIC27 KOs in four cell lines; HEK293, HepG2, HeLa and HAP1 cell lines which we immunoblotted using anti-MIC27 antibody as shown in S3 Fig. A-D. 

2) I do not find the Mic27 RNAi silencing to be very convincing in Figure 1. The band appears to be at the same level as the control on the very left. No quantification of Westerns from this silencing are shown. Given the authors later utilize the Mic27 KO cell lines to convincingly demonstrate that the 55 kDa band is not derived from Mic27 either, the RNAi-silencing of Mic27 does not seem so relevant here. I would suggest the authors remove this experiment from the manuscript, keeping the robust Mic26 RNAi.

Response: Thank you for the comment. We have now removed the last lane in Fig. 1A (showing the MIC27 RNAi). Accordingly, the text in the results and materials and methods has been removed in the manuscript.

3) Regarding the experiments to remove the serine residues that represent putative O-linked glycosylation sites. As far as I know, this is not the only amino acid that can undergo this modification. Furthermore, since there is no evidence that the 22 kDa form is glycosylated, the authors should make clear these sites are theoretical and also other amino acids with exposed hydroxyl groups in their side chains can also be I n theory glycosylated. But once again, this is more for transparency. The authors have shown convincingly that the 55 kDa glycosylated protein is not derived from either Mic26 or Mic27.

Response: This is a good point. We have added the word in silico/‘Theoretical’ in the text in Lines 380, 383 and 416 (Figure 3 legend).

4) On lines 26-27, the authors state that MICOS is a well-conserved complex. I disagree with the phrasing. Mic60 and Mic10 are well-conserved subunits and the role in making contact sites appears to be conserved in land plants (Michaud et al., 2016 Curr. Biol. 26:627-639), trypanosomes (Kaurov et al., 2018 Curr. Biol. 28:3393-3407) and alphaproteobacteria (Munoz-Gomez et al., bioRxiv, 2022.06. 14.496148). Furthermore, the trypanosome MICOS subunit composition is very different from yeast and human MICOS. This is consisent with most of the other MICOS subunits from animals/yeast not distributed in the eukaryotic tree of life in contast to Mic10 and Mic60. This statement should thus be modified to reflect how MICOS as adopted different subunit compositions.

Response: Thank you for the insight. We accordingly changed the text in the manuscript. The modified text is present in lines 29-37.

5) On line 236, please cite the already cited Munoz-Gomez et al. (2015) and Huynen et al. (2016) articles, as they show by molecular phyogenetics that human Mic26 and Mic27 are paralogs.

Response: According to the reviewer #2 point 2, we have removed the MIC27 downregulation data and therefore the need to explain that MIC26 and MIC27 are paralogs did not arise in this section of the revised version. However, MIC26 and MIC27 are paralogs is mentioned in lines 36-37.

6) Line 390, replace ‘apparently’ with ‘putative’ as the latter seems convey the meaning better.

Response: According to the reviewer’s comment, we have replaced the word ‘apparently’ with ‘putative in line 432’.

7) Line 498, delete ‘emphatically’ or replace with another word like ‘convincingly’.

Response: The word ‘Convincingly’ has been used in the revised manuscript in line 544.

References:

1. Koob S, Barrera M, Anand R, Reichert AS. The non-glycosylated isoform of MIC26 is a constituent of the mammalian MICOS complex and promotes formation of crista junctions. Biochimica et biophysica acta. 2015;1853(7):1551-63.

2. Weber TA, Koob S, Heide H, Wittig I, Head B, van der Bliek A, et al. APOOL is a cardiolipin-binding constituent of the Mitofilin/MINOS protein complex determining cristae morphology in mammalian mitochondria. PloS one. 2013;8(5):e63683.

3. Anand R, Kondadi AK, Meisterknecht J, Golombek M, Nortmann O, Riedel J, et al. MIC26 and MIC27 cooperate to regulate cardiolipin levels and the landscape of OXPHOS complexes. Life Sci Alliance. 2020;3(10):e202000711.

4. Rueter J, Rimbach G, Huebbe P. Functional diversity of apolipoprotein E: from subcellular localization to mitochondrial function. Cellular and molecular life sciences : CMLS. 2022;79(9):499.

5. Huang X, Zhang KJ, Jiang JJ, Jiang SY, Lin JB, Lou YJ. Identification of Crucial Genes and Key Functions in Type 2 Diabetic Hearts by Bioinformatic Analysis. Front Endocrinol (Lausanne). 2022;13:801260.

---

## [Decision Letter · Decision Letter 1]

17 May 2023

PONE-D-22-33726R1MIC26 and MIC27 are bona fide subunits of the MICOS complex in mitochondria and do not exist as glycosylated apolipoproteinsPLOS ONE

Dear Dr. Kondadi,

Thank you for submitting your manuscript to PLOS ONE. After careful consideration, we feel that it has merit but does not fully meet PLOS ONE’s publication criteria as it currently stands. Therefore, we invite you to submit a revised version of the manuscript that addresses the points raised during the review process.

I am inclined to accept this manuscript after the authors carry out the minor revision and reply to the minor points raised by Reviewer 3.

We look forward to receiving your revised manuscript.

Kind regards,

Benedetta Ruzzenente

Academic Editor

PLOS ONE

Journal Requirements:

Reviewers' comments:

Reviewer's Responses to Questions

**Comments to the Author**

1. If the authors have adequately addressed your comments raised in a previous round of review and you feel that this manuscript is now acceptable for publication, you may indicate that here to bypass the “Comments to the Author” section, enter your conflict of interest statement in the “Confidential to Editor” section, and submit your "Accept" recommendation.

Reviewer #1: All comments have been addressed

Reviewer #3: All comments have been addressed

2. Is the manuscript technically sound, and do the data support the conclusions?

Reviewer #1: Yes

Reviewer #3: Yes

3. Has the statistical analysis been performed appropriately and rigorously? 

Reviewer #1: N/A

Reviewer #3: Yes

4. Have the authors made all data underlying the findings in their manuscript fully available?

Reviewer #1: Yes

Reviewer #3: Yes

5. Is the manuscript presented in an intelligible fashion and written in standard English?

Reviewer #1: Yes

Reviewer #3: Yes

6. Review Comments to the Author

Reviewer #1: (No Response)

Reviewer #3: I am satisfied with the authors' responses to my review. However, in responding to a couple of the points, two very minor point arose, which I ask the authors to address prior to acceptance. I also include some suggestions, that the authors can consider for clarification. Once these small issues in the text are addressed, the manuscript will be suitable for acceptance. Quoted line numbers correspond to unmarked version.

Minor points (mandatory)

1) Lines 31-33. Mic60 is conserved from alphaproteobacteria to virtually all eukaryotes, whereas Mic10 is widely distributed in the later domain of life. Please rephrase this sentence accordingly.

2) Line 365: ‘…various predicted…”

Suggestions (up to authors to address):

3) Line 394: “protein that is not apparently glycolsyated.”

4) Line 616: Rather use full form of w.r.t. abbreviation

50Figures 1 and S2. Point out correct 22 kDa band, incorrect 55kDa band and other nonspecific bands.

7. PLOS authors have the option to publish the peer review history of their article (what does this mean?). If published, this will include your full peer review and any attached files.

Reviewer #1: No

Reviewer #3: No

---

## [Author Response · Author response to Decision Letter 1]

18 May 2023

18th May 2023, Lubeck et al 2023, PLOS One (Minor Revision)

Reviewer #3: I am satisfied with the authors' responses to my review. However, in responding to a couple of the points, two very minor point arose, which I ask the authors to address prior to acceptance. I also include some suggestions, that the authors can consider for clarification. Once these small issues in the text are addressed, the manuscript will be suitable for acceptance. Quoted line numbers correspond to unmarked version.

Minor points (mandatory)

1. Lines 31-33. Mic60 is conserved from alphaproteobacteria to virtually all eukaryotes, whereas Mic10 is widely distributed in the later domain of life. Please rephrase this sentence accordingly.

Response: Thank you for the constructive comment. We have changed the text accordingly.

2. Line 365: ‘…various predicted…”

Response: Thank you. We have changed the text accordingly.

Suggestions (up to authors to address):

3. Line 394: “protein that is not apparently glycolsyated.”

Response: Thank you. We have changed the text accordingly.

4. Line 616: Rather use full form of w.r.t. abbreviation

Response: Thank you. We have changed the text accordingly.

5. Figures 1 and S2. Point out correct 22 kDa band, incorrect 55kDa band and other nonspecific bands.

Response: According to the reviewer, this is an optional response from our side. We believe that during the course of the manuscript, it is abundantly clear which protein bands are specific and nonspecific. Therefore, the original figure can stay the same as after major round of revision.

28th March 2023, Lubeck et al 2023, PLOS One (Major Revision)

Review Comments to the Author

Reviewer #1: In their manuscript submitted to PLoS ONE, Lubeck et al. investigate the 55 kDa form of the MICOS component MIC26 that was initially reported to be a glycosylated, secreted version of the 22 kDa protein. This protein was described to be present in lipoproteins, leading to the categorization of MIC26 as apolipoprotein O (APOO) and, subsequently, of the related MICOS subunit MIC27 as apolipoprotein O-like (APOOL). Biochemical studies of the two proteins have focused on their function in mitochondria as part of MICOS, and the function of the 55 kDa protein has remained unclear.

Surprisingly, the authors demonstrate conclusively that this mysterious protein is not a derivate of MIC26 or of MIC27, and instead represents an unrelated protein that is crossreactive with certain anti-MIC26 antibodies. Moreover, they report that there is no specific secreted species of MIC26 in cell culture supernatant or in mouse plasma, which agrees with analyses of secretomes. These findings suggest that MIC26 is localized only within the cell, in line with its well-established mitochondrial function.

This study is an important correction to the body of MICOS literature and will be crucial to ongoing research focusing on the 55 kDa species, an unidentified protein with possibly important functions.

Some points that should be addressed:

1. The mouse plasma Western blots in Fig. 4C, D are empty. As positive controls, authentic MIC26 from cell extracts as well as an established plasma protein need to be included on the same gel.

Response: Thank you for the constructive comment. We have now updated the WBs in Fig. 4C and 4D by using HEK293 cell lysates, apart from murine plasma samples, which were immunoblotted using anti-MIC26 antibodies #1 and #2. We used an antibody against albumin as a positive control in WBs using murine plasma which were added during revision. 

In addition, we also updated the S5 Fig.C and S5 Fig.D (previously S3 Fig.C & D), where we performed WBs (using two additional antibodies) using anti-MIC26 #3 and #4 antibodies not only in murine plasma but also in HEK293 lysates as in Fig.4. Anti-albumin antibody was used a positive control for murine plasma.

2. The authors discuss the inefficient deglycosylation of the 55 kDa protein in the original study (Lamant et al., 2006), but not that they themselves have reported quite efficient conversion of the 55 kDa band into what they thought was deglycosylated MIC26 (Koob et al., 2015, BBA). How do the authors explain this? Presumably the lower MW band that appears after treatment is the unglycosylated version of the 55 kDa protein = not MIC26 but still recognized by the MIC26 antiserum. The authors should provide evidence in an experiment analogous to their published one, that the deglycosylated protein is not recognized by more specific antisera, and that the assay yields the same results in MIC26 deletion cells.

Response: Thank you for the suggestion. We have performed experiments with WT and MIC26 KOs in HEK293 cells where the cell lysates were treated with or without chondroitinase ABC (cABC) enzyme and probed with anti-MIC26 antibody #1. We observe a slight reduction in the 55 kDa band, while we do not observe an accumulation in the 22 kDa region for WT as well as MIC26 KOs upon cABC treatment. It suggests that the accumulation (shown in previous manuscript) is only found probably under certain selective conditions. In fact, even after repetitive experiments, we were not able to observe the result which was shown before. Therefore, our observations in previous manuscript could be explained because of nonspecific proteolytic cleavage under specific conditions. 

As a positive control for proper cABC enzymatic digestion upon our treatment, we probed the same cell lysates with an anti-Chondroitin-4-Sulfate antibody (Sigma Aldrich, MAB2030, 1:1000). A clear accumulation of released chondroitin-4-sulfate proteoglycans can be observed in WT and MIC26 KO cells treated with cABC (shown in asterisks). 

3. Since most of the original criteria for the inclusion of MIC26 into the apolipoprotein family have been disproven in this study, the authors should discuss whether the last claim from the original study, sequence similarity to apolipoproteins, has any merit and may be related to lipid binding by MIC26 or MIC27. This issue, as well as the question whether or not their classification as APOO and APOOL should be rejected, is of continued relevance to the field.

Response: According to our observations in this manuscript, we have only shown that the 55 kDa protein is nonspecifically recognised by certain anti-MIC26 antibodies. Therefore, MIC26 does not exist as a modified 55 kDa protein. MIC26 (1) and MIC27 (2) sequence analysis have shown that they possess apolipoprotein family domains. Accordingly, MIC27 has been shown to bind to cardiolipin species. Previously, we have also shown that double knockouts of MIC26 and MIC27 led to reduced cardiolipin levels (3). Therefore, while we argue that the glycosylated version of MIC26 does not exist, we think that MIC26 and MIC27 are still apolipoproteins whose function is not completely understood in the mitochondria. Another human apolipoprotein, ApoE is also known to localise to mitochondria (apart from other organelles) and plays an important role in cholesterol metabolism in addition to its secretion in plasma (4). Altogether, MIC26 and MIC27 are mitochondrial apolipoproteins, although MIC26 does not exist as a secreted glycosylated protein. 

4. It would be helpful if the authors performed a subcellular fractionation to corroborate with different antibodies that MIC26 does not enter the secretory pathway.

Response: This is a very helpful suggestion from the reviewer. We have performed subcellular fractionation assays, where mitochondrial fraction was separated from the combined ER/Golgi/cytosolic fraction, using HEK293 cell lysates. Western blotting using anti-ANT2 antibody showed immunopositivity in total and mitochondrial fractions as expected while calreticulin was found in the ER/Golgi/cytosol fraction as expected. Using all four anti-MIC26 antibodies, we found that mitochondrial MIC26 was exclusively present in total and mitochondrial fractions (and not in ER/Golgi/cytosolic fraction) corroborating that MIC26 does not enter the secretory pathway. The new figure is included in the revised version as S2 Figs. A-D. 

Reviewer #2: This paper analyzes the 55 kDa version of the Mic26 protein that has been described to be a glycosylated Mic26 in some publications. Using knockout and different antibodies, as well as mutagenesis of glycosylation sites, the expression of the GFP-tagged protein, and mass spectrometry, the authors show that the observed 55 kDa band is not the glysosylated Mic26, but most likely an antibody-specific cross-reactive band.

In my opinion, the paper successfully proves the point and is an important contribution to the field, because it shows that Mic26 exists only as mitochondria-associated protein. The question remains, however, what protein was identified in the serum using ApoO antibody (Lamant et al, JBC, 2006) and why this antibody did not show the presence of the 22 kDa form of ApoO in the cell lysate. However, it is important to present convincing data that strongly disprove that the observed 55 kDa band was ApoO and open the previously published results to discussion. For that reason, I support the publication of this study.

I have only minor comments to the paper:

- The authors show that there is no increase in protein levels of Mic26/ApoO in their model of diabetic mouse. Because these data partially contradict the previously published information about the upregulation of the Mic26 transcription in diabetes, the authors should check the transcription of Mic26 in their mouse model and show this data in addition to the western blot data.

Response: We thank the reviewer for this constructive criticism. We have used liver tissues from db/db.BKS diabetic mice model and compared them with db/+ control mice for checking the levels of MIC26 transcripts. Our data in Fig. 5E shows that MIC26 as well as MIC27 transcripts are similar in diabetic mice compared to control mice 

In addition to this, we checked the transcriptomics data from Huang et al., 2022 (5), where they performed transcriptome sequencing from heart tissue of BKS wildtype control and db/db.BKS mouse. Briefly, we retrieved the “geneCounts” and “geneFPKM” files from NCBI GEO database (GSE161931) and performed DESeq2 analysis on the count file. We did not observe any significant difference in Fragments Per Kilobase of transcript per Million mapped reads (FPKM) of MIC26 and MIC27 between BKS wildtype control and db/db.BKS mouse. DESeq2 analysis revealed a minor but not significant decrease in log2 Fold Change of MIC26 and a minor increase in MIC27. Considering the transcriptomics data, we did not notice any considerable difference in mRNA expression of MIC26 and MIC27 in db/db.BKS mouse as compared to db/+ wildtype control strain.

Would the authors venture to propose what the origin of the 55 kDa band is? An immunoprecipitation with the antibody that recognizes the 55 kDa band might give an answer to this and explain the observations from the Lamant et al paper.

Response: Thank you for the comment. Initially, we were of a similar opinion as the reviewer #2. However, we think that various MIC26 antibodies recognise different proteins which is clear upon careful observation of the blots in Fig 1B-D. In Fig. 1B, the detected intracellular as well as secreted protein in WT and MIC26 KO HepG2 cells runs slightly below 55 kDa marker while the proteins recognised using antibodies #2 (Fig. 1C) and #3 (Fig.1D) run above 55 kDa marker. Therefore, antibody #1 is not recognising the same as antibody #2 and #3. Further, it is not possible that antibodies # 2 and # 3 are recognising the same protein as antibody #2 is not recognising any secreted protein while antibody #3 is recognising secreted proteins above 55 kDa marker band. Thus, due to our observations that multiple proteins are recognised with various antibodies, we were not encouraged to investigate the nature of the 55 kDa protein further. We also think that investigating the nature of these protein/s is beyond the scope of this manuscript.

Text correction: Line 572, N-Terminus should be N-terminus.

Response: The text has been changed accordingly in the new line 619.

Reviewer #3: The aim of the manuscript by Lubeck at al. is to resolve once and for all the false notion that the apoplipoprotein subunit of human MICOS Mic26 exists as two isoforms: one bona fide subunit of the MICOS complex as a 22 kDa form and a secreted, glycosylated 55 kDa form. This very thorough study uses a battery of KO human cell lines, expression of tagged forms, putative glycosylation mutants, antibodies and RNAi knockdowns to support their conclusion that the 55 kDa form detected using one of the Mic26 antibodies (antibody #1) is not derived from the 22 kDa form. Furthermore, they eliminate the possibility that this 55 kDa band can arise from the Mic27 paralog. Finally, they show that levels of Mic26 protein is not affected in the diabetes mouse model, thanks to their efforts to show that antibodies #1 and #2 raised against the human ortholog crossreacts with the mouse; this result contradicts transcriptomic data claiming Mic26 mRNA is affected in these mice. The manuscript is well written and presents novel findings that are appropriate for publication in PLoS One. The presented data overall supports the conclusions made in the paper. Thus, I recommend the manuscript be accepted after minor revision. I list below some minor points that I think should be addressed by the authors to finalize the manuscript.

1) The 55 kDa region of the Western blot probed with the Mic27 antibody is shown in Figures 1 and 2. I am sure the authors are showing this region, but it would be more convincing if they also show at least an exemplar whole blot image, as they have done for the various Mic26 antibodies.

Response: Thank you for the comment. We have now included whole Western blots of WT, MIC26 and MIC27 KOs in four cell lines; HEK293, HepG2, HeLa and HAP1 cell lines which we immunoblotted using anti-MIC27 antibody as shown in S3 Fig. A-D. 

2) I do not find the Mic27 RNAi silencing to be very convincing in Figure 1. The band appears to be at the same level as the control on the very left. No quantification of Westerns from this silencing are shown. Given the authors later utilize the Mic27 KO cell lines to convincingly demonstrate that the 55 kDa band is not derived from Mic27 either, the RNAi-silencing of Mic27 does not seem so relevant here. I would suggest the authors remove this experiment from the manuscript, keeping the robust Mic26 RNAi.

Response: Thank you for the comment. We have now removed the last lane in Fig. 1A (showing the MIC27 RNAi). Accordingly, the text in the results and materials and methods has been removed in the manuscript.

3) Regarding the experiments to remove the serine residues that represent putative O-linked glycosylation sites. As far as I know, this is not the only amino acid that can undergo this modification. Furthermore, since there is no evidence that the 22 kDa form is glycosylated, the authors should make clear these sites are theoretical and also other amino acids with exposed hydroxyl groups in their side chains can also be I n theory glycosylated. But once again, this is more for transparency. The authors have shown convincingly that the 55 kDa glycosylated protein is not derived from either Mic26 or Mic27.

Response: This is a good point. We have added the word in silico/‘Theoretical’ in the text in Lines 380, 383 and 416 (Figure 3 legend).

4) On lines 26-27, the authors state that MICOS is a well-conserved complex. I disagree with the phrasing. Mic60 and Mic10 are well-conserved subunits and the role in making contact sites appears to be conserved in land plants (Michaud et al., 2016 Curr. Biol. 26:627-639), trypanosomes (Kaurov et al., 2018 Curr. Biol. 28:3393-3407) and alphaproteobacteria (Munoz-Gomez et al., bioRxiv, 2022.06. 14.496148). Furthermore, the trypanosome MICOS subunit composition is very different from yeast and human MICOS. This is consisent with most of the other MICOS subunits from animals/yeast not distributed in the eukaryotic tree of life in contast to Mic10 and Mic60. This statement should thus be modified to reflect how MICOS as adopted different subunit compositions.

Response: Thank you for the insight. We accordingly changed the text in the manuscript. The modified text is present in lines 29-37.

5) On line 236, please cite the already cited Munoz-Gomez et al. (2015) and Huynen et al. (2016) articles, as they show by molecular phyogenetics that human Mic26 and Mic27 are paralogs.

Response: According to the reviewer #2 point 2, we have removed the MIC27 downregulation data and therefore the need to explain that MIC26 and MIC27 are paralogs did not arise in this section of the revised version. However, MIC26 and MIC27 are paralogs is mentioned in lines 36-37.

6) Line 390, replace ‘apparently’ with ‘putative’ as the latter seems convey the meaning better.

Response: According to the reviewer’s comment, we have replaced the word ‘apparently’ with ‘putative in line 432’.

7) Line 498, delete ‘emphatically’ or replace with another word like ‘convincingly’.

Response: The word ‘Convincingly’ has been used in the revised manuscript in line 544.

References:

1. Koob S, Barrera M, Anand R, Reichert AS. The non-glycosylated isoform of MIC26 is a constituent of the mammalian MICOS complex and promotes formation of crista junctions. Biochimica et biophysica acta. 2015;1853(7):1551-63.

2. Weber TA, Koob S, Heide H, Wittig I, Head B, van der Bliek A, et al. APOOL is a cardiolipin-binding constituent of the Mitofilin/MINOS protein complex determining cristae morphology in mammalian mitochondria. PloS one. 2013;8(5):e63683.

3. Anand R, Kondadi AK, Meisterknecht J, Golombek M, Nortmann O, Riedel J, et al. MIC26 and MIC27 cooperate to regulate cardiolipin levels and the landscape of OXPHOS complexes. Life Sci Alliance. 2020;3(10):e202000711.

4. Rueter J, Rimbach G, Huebbe P. Functional diversity of apolipoprotein E: from subcellular localization to mitochondrial function. Cellular and molecular life sciences : CMLS. 2022;79(9):499.

5. Huang X, Zhang KJ, Jiang JJ, Jiang SY, Lin JB, Lou YJ. Identification of Crucial Genes and Key Functions in Type 2 Diabetic Hearts by Bioinformatic Analysis. Front Endocrinol (Lausanne). 2022;13:801260.

---

## [Editor Report · Decision Letter 2]

23 May 2023

MIC26 and MIC27 are bona fide subunits of the MICOS complex in mitochondria and do not exist as glycosylated apolipoproteins

PONE-D-22-33726R2

Dear Dr. Kondadi,

We’re pleased to inform you that your manuscript has been judged scientifically suitable for publication and will be formally accepted for publication once it meets all outstanding technical requirements.

Kind regards,

Benedetta Ruzzenente

Academic Editor

PLOS ONE
---

## [Editor Report · Acceptance letter]

29 May 2023

PONE-D-22-33726R2 

MIC26 and MIC27 are bona fide subunits of the MICOS complex in mitochondria and do not exist as glycosylated apolipoproteins 

Dear Dr. Kondadi:

I'm pleased to inform you that your manuscript has been deemed suitable for publication in PLOS ONE. Congratulations! Your manuscript is now with our production department. 

Kind regards, 

on behalf of

Dr. Benedetta Ruzzenente 

Academic Editor

PLOS ONE